# Exploring Accurate and Transparent Domain Adaptation in Predictive Healthcare via Concept-Grounded Orthogonal Inference

**Pengfei Hu**[1]  **Chang Lu**[1]  **Feifan Liu**[2]  **Yue Ning**[1]

## Abstract

Deep learning models for clinical event prediction on electronic health records (EHR) often suffer performance degradation when deployed under different data distributions. While domain adaptation (DA) methods can mitigate such shifts, their "black-box" nature prevents widespread adoption in clinical practice where transparency is essential for trust and safety. We propose `ExtraCare` to decompose patient representations into invariant and covariant components. By supervising these two components and enforcing their orthogonality during training, our model preserves label information while exposing domain-specific variation at the same time for more accurate predictions than most feature alignment models. More importantly, it offers human-understandable explanations by mapping sparse latent dimensions to medical concepts and quantifying their contributions via targeted ablations. `ExtraCare` is evaluated on two real-world EHR datasets across multiple domain partition settings, demonstrating superior performance along with enhanced transparency, as evidenced by its accurate predictions and explanations from extensive case studies.

## 1. Introduction

Health equity requires that AI healthcare systems perform robustly and fairly across diverse patient populations (Braveman, 2006; Starfield et al., 2012). Although deep learning techniques have shown potential in supporting predictive healthcare by learning longitudinal patient data from electronic health records (EHR), there is evidence that a model trained on one patient cohort often does not work well for another patient population (Pfohl et al., 2019; Guo et al., 2022; Yang et al., 2023a). This concern is especially acute in clinical settings due to different devices, clinical standards, and evolving environments across clinical institutes. Such discrepancy induces distribution shifts (covariates) between training data (source domain) and test data (target domain), leading to degraded predictive performance when deployed in unseen environments. As such, domain adaptation (DA) methods are motivated to help models adapt across domains.

Despite the promise of clinical DA, it is rarely embraced by clinicians in routine practice due to transparency concerns (Chaddad et al., 2023). In high-stakes settings, explainability is usually emphasized as a prerequisite for acceptance, accountability, and safe integration into clinical workflows (Duell et al., 2021; Abbas et al., 2025; Han et al., 2025). Yet, most DA solutions are not designed to be interpretable, since few of them operate on clinical entities beyond hidden representations. This gap matters because DA is a selection process over patient representations (Xie et al., 2022). Understanding the medical meaning of what being kept (**invariant**) and removed (**covariant**) is critical to clinicians. Without such transparency, DA risks becoming an opaque procedure that is difficult to validate and debug when failures occur.

Although research focusing on interpretable DA remains limited, there are precedents in other related clinical tasks that explicitly connect learned representations to medical concepts (Vu et al., 2020; Liu et al., 2022). Among them, AutoCodeDL (Wu et al., 2024) explores the sparse autoencoder (SAE) mapping token-level embeddings to medical codes, enabling a concept-level interpretation of model behavior. This idea is highly relevant to healthcare event prediction because many predictive tasks are inherently code-level prediction problems (e.g., diagnosis prediction), where outputs correspond to standardized clinical codes. If a model's latent patient representation can be decomposed into a sparse set of interpretable factors aligned with medical concepts, then clinicians can potentially inspect which concepts drive predictions, how strongly they contribute, and whether they are clinically plausible. Therefore, sparse vectors serve as a compelling building block for DA models to provide explanations grounded in clinical semantics.

[1]Department of Computer Science, Stevens Institute of Technology, Hoboken, NJ, United States [2]UMass Chan Medical School, University of Massachusetts Amherst, Amherst, MA, United States. Correspondence to: Yue Ning <yue.ning@stevens.edu>.

*Proceedings of the 43rd International Conference on Machine Learning*, Seoul, South Korea. PMLR 306, 2026. Copyright 2026 by the author(s).

However, applying SAEs to clinical DA problems is not straightforward: (1) Sparse models are often used post hoc (Cunningham et al., 2023), which yields biased explanations that are more reflective of model parameters than of patient records by decoupling the reconstruction from the training objective; (2) Most DA approaches emphasize learning invariant features with removal of covariate signals, making explanations remain incomplete: we may learn what concepts are retained, but we still cannot see what is removed as domain-dependent. In clinical practice, getting accurate explanations and understanding what the model ignores are crucial to ensure meaningful subgroup patterns are not erased during adaptation.

To this end, we introduce ExtraCare (**Ex**ploring accurate and **Tra**nsparent solutions in health**Care** applications), an orthogonal and concept-grounded inference framework. Building upon SAE, we leverage the sparse vectors to reconstruct aligned features in the latent space, and an orthogonal inference module is then developed to factorize patient representations into two components: domain covariates and label information. Next, we impose distinct supervisions on both components during training so that both vectors can capture invariant and covariant features by enforcing their orthogonality. By ablating sparse activations and examining the induced changes in model outputs, we attribute each non-zero dimension to medical concepts (e.g. groups of conditions), and further distinguish whether these concepts primarily drive label prediction or reflect domain-specific variation. Our contributions can be summarized as follows:

- To our knowledge, we are one of the first to explore SAE-driven orthogonal inference for domain adaptation in health event prediction, and we also provide theoretical justification for the induced orthogonality.

- Our method offers a prevalence-aware interpretation of patient representations, enabling clinicians to audit, contest, and reason about the adaptation behavior rather than treating it as a black box.

We evaluate our models on two real-world EHR datasets and demonstrate superiority over state-of-the-art DA baselines, indicating that our model can provide explanations without sacrificing predictive performance.

## 2. Related Work

**Domain Adaptation (DA).** The common purpose of DA solutions is to reduce the distributional discrepancy between source and target domains (Farahani et al., 2021). Current DA solutions can be summarized into two mainstream categories (Zhao et al., 2020). Self-training leverages unlabeled data by training either on pseudo-labels generated by a source model or under consistency regularization (Liu

et al., 2021; Sun et al., 2022a). In this work, we focus on the feature adaptation paradigm, which captures domain divergence by aligning different domain data into a shared space via the customized objective, like mean discrepancy (Gretton et al., 2006; Yan et al., 2017; Wang et al., 2023) and adversarial training losses (Ganin et al., 2016; Zhang et al., 2019; Ge et al., 2023). Within this scope, a complementary research line is to disentangle learned representations. These methods typically assume linear dependency in the feature space and then each extracted feature can be split into a domain-invariant and a domain-specific components. Domain separation network (Bousmalis et al., 2016) and follow-up works partition latent features into private (domain-specific) and shared subspaces by using similar orthogonality constraints (Sun et al., 2022b; Liu et al., 2025), which also tightens theoretical bounds on target performance by retaining crucial domain-specific cues.

**Domain Adaptation in Clinical Applications.** Domain adaptation becomes even more crucial in predictive healthcare, since models are usually trained on data from different hospitals, institutions, or time periods. Aligning patient-level distributions across hospitals is now effective in reducing performance drops when adapting predictive models to a new site (Guo et al., 2022; Wang et al., 2024); transfer learning also internalizes shared patterns across different sources (Gupta et al., 2020). Similarly, these techniques are widely applied to medical image analysis (Feng et al., 2023; Tong et al., 2025b; Chen et al., 2024; Zhu et al., 2024): models trained on one scanner often underperform on another due to different imaging devices. Domain adaptation is thus seen as a key tool in building robust clinical AI that works in diverse settings, such as multi-center disease diagnosis (Zhu et al., 2025). However, most of these approaches still treat the learning phase as a black-box optimization by focusing on accuracy and robustness, which leaves a gap in interpretability and trustworthiness.

**Transparency in Clinical Prediction.** Post-hoc attribution (e.g. SHAP and LIME) and integrated gradient methods are used to get feature-level contributions on the final output (Payrovnaziri et al., 2020; Duell et al., 2021). However, these explanations are sensitive to method settings and may not reflect meaningful reasoning (Zhou et al., 2025; Lin et al., 2026). Attention scores are also used for explaining outputs, as their dynamic visual patterns can reveal how a model allocates diagnostic focus (Lu et al., 2021; Hu et al., 2026b; Zhu et al., 2026). In many clinical natural language processing (NLP) tasks, label-wise attention highlights words deemed relevant to each label in medical coding tasks (Vu et al., 2020; Liu et al., 2022; Abbas et al., 2025). Building upon this, recent work further enhances transparency upon sparse vectors in medical coding (Wu et al., 2024), which builds a human-understandable dictionary between representations and clinical concepts. More

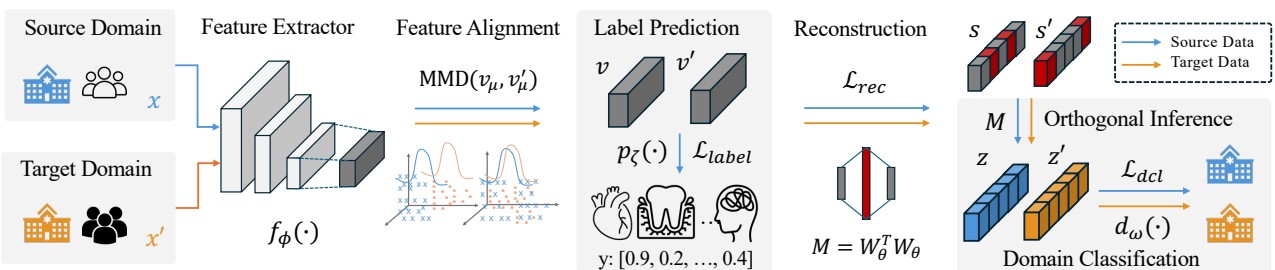

*Figure 1.* **Overview of `ExtraCare` architecture for clinical domain adaptation problems.** Inputs $x, x'$ are encoded by $f_\phi(\cdot)$ into $v, v'$, supervised for label prediction with $p\zeta(\cdot)$. A sparse autoencoder $h_\theta(\cdot)$ induces a dictionary metric $M = W_\theta^\top W_\theta$ and enables orthogonal inference that factorizes representations into invariant features and domain-specific residuals $z$, supervised by a domain classifier $d_\omega(\cdot)$.

importantly, this paradigm can also generalize to a broad range of code-level prediction tasks, aligning well with clinical DA, where most problems follow this formulation. However, directly applying such mappings remains challenging, since models are usually required to explain the adaptation process not just the final output. Thus, we aim to overcome these issues in this paper.

## 3. Methodology

### 3.1. Problem Definition

Given an EHR dataset $\mathcal{D} = \{(x_i, y_i)\}_{i=1}^n$ with $n$ patients in total, the $i$-th patient's data $X_i$ consists of a longitudinal sequence of visits $\{V_1^{(i)}, V_2^{(i)}, \ldots, V_T^{(i)}\}$ spanning $T$ admission records. Each visit involves various medical concepts (codes) from the vocabulary $\mathcal{C}$, and single patient data is the sequence of encoded vectors $x_i \in \mathbb{R}^{T \times |\mathcal{C}|}$. Existing clinical prediction works typically train an encoder $f_\phi(\cdot)$ followed by a label predictor $p_\zeta(\cdot)$ for predicting the occurrence of future healthcare events $y \in \mathbb{R}^o$ (e.g. disease codes), where the label dimension $o$ depends on specific tasks.

To transfer trained $f_\phi(\cdot)$ across patient populations, $\mathcal{D}$ is divided into a source training domain $\mathcal{D}_s = \{(x_i, y_i)\}_{i=1}^{n_s}$ of $n_s$ labeled samples and a target test domain $\mathcal{D}_t = \{x_i'\}_{i=1}^{n_t}$ of $n_t$ unlabeled samples. Both domains are usually sampled from similar but not identical probability distributions $P_s$ and $P_t$, and the expected target risk $\mathbb{E}_{P_t}[\ell(\phi; x, y)]$ with defined loss $\ell(\cdot)$ can be bounded by leveraging the source labeled data. Concretely, *we assume that $\mathbb{E}_{P_s}[y|x = x] = \mathbb{E}_{P_t}[y|x = x']$ in light of the available data, such that covariate shift is the only thing that changes between $\mathcal{D}_s$ and $\mathcal{D}_t$.* Most domain adaptation studies approximate the expectation over $P_t$ via samples from $P_s$:

$$\mathbb{E}_{P_t}[\ell(\phi, \zeta; x', y')] \triangleq \mathbb{E}_{P_s}[w(x)\,\ell(\phi, \zeta; x, y)], \quad (1)$$

where $w(x)$ represents the density ratio between the marginal distribution of inputs in the target and that in the source domains, and the loss $\ell(\cdot)$ can be specified on detailed tasks. A notation table is provided in Appendix G.

### 3.2. Feature Extraction & Alignment

Before adapting patient features across domains, the first step is to feed the input sample pair $(x, x')$ into a learnable feature extractor (encoder) $f_\phi(\cdot) : x \mapsto v$ with parameter $\phi$ to get patient-specific representations $v, v'$:

$$v = f_\phi(x), \quad v' = f_\phi(x'), \quad (2)$$

where $x \in \mathcal{D}_s$ and $x' \in \mathcal{D}_t$. Note that, $f_\phi(\cdot)$ could be any deep learning backbone, including RNN (Chung et al., 2014), GNN (Kipf, 2016), or Transformer (Vaswani et al., 2017), and we can specify the choices of feature encoder networks based on the applications.

Unlike regular predictive settings, the encoded features $v, v'$ cannot be directly used for label prediction, since they include both label information and domain information (i.e., patient covariates). The former is invariant across domains, whereas the latter may introduce spurious correlations and can harm generalization. To help $f_\phi(\cdot)$ retain only label-related features, prior studies (Li et al., 2018; Kang et al., 2022) have shown that reducing the distributional divergence between $P_s$ and $P_t$ helps models focus on domain-shared components, which can be viewed as an approximation of the invariant factors.

**Objective 1: Supervised Loss for Label Prediction.** Regularization is commonly used upon the loss function to close the distribution gap. Accordingly, we adopt the Maximum Mean Discrepancy (MMD) (Muandet et al., 2013; Long et al., 2015) to align the embedding space of $v$ and $v'$. Note that, the label supervision upon encoded features should be also involved. With labeled samples $(x, y)$ in $\mathcal{D}_s$, we consider the cross-entropy loss $\ell_{\text{CE}}(\cdot)$, so that parameters $\phi$ and $\zeta$ can be jointly optimized by

$$\mathcal{L}_{\text{label}} = \mathbb{E}_{P_s}[\ell_{\text{CE}}(\phi, \zeta; x, y)] + \lambda_1 \frac{\text{MMD}(v_\mu', v_\mu)}{\|\text{sg}(v_\mu)\|_{\mathcal{F}}^2}, \quad (3)$$

where $v_\mu', v_\mu$ are averaged over the data batch, and $\|\cdot\|_{\mathcal{F}}$ denotes the norm $\mathcal{F}$ induced by the inner product. The discrepancy is rescaled by $\text{sg}(v_\mu)$ with stopped gradient

back-propagation, and $\lambda_1$ controls the regularization weight. Compared to regular predictive frameworks, the regularization is used to remove the domain-specific component from $v$ and $v'$, which are close to their domain-invariant parts.

### 3.3. Reconstruction of Aligned Features

Recall that our goal is to explain adaptation by mapping medical concepts to hidden features, we employ a sparse autoencoder (SAE) to reconstruct $v$ from a sparse feature $s$, which encourages each dimension in $s$ to capture a distinct and semantically meaningful factor for interpretation (Cunningham et al., 2023; Kissane et al., 2024):

$$s = h_\theta(v) = \mathrm{ReLU}(W_\theta v), \quad \hat{v} = W_\theta^\top s. \qquad (4)$$

Here, $s \in \mathbb{R}^{d_s}$ represents the sparse vector, and tied weights $W_\theta, W_\theta^\top$ are used for the whole reconstruction. $\mathrm{ReLU}(\cdot)$ denotes the activation function, and weight bias terms are omitted for both the encoder and decoder.

However, it is important to note that $v$ is not exactly domain-invariant when optimized only with label supervision. As discussed in Section 3.2, invariant features should be both *predictive of the outcome* and *insensitive to domain information*. This motivates us to explicitly disentangle $v$ into invariant and covariant features with distinct objectives:

- Invariant feature is used only for label prediction;

- Covariate feature is used to classify domains only.

To factorize encoded features, prior works have shown that these two subspaces can be orthogonal (linear-irrelevant) in the vector space (Shen et al., 2022; Yang et al., 2023a). Inspired by this, we further adjust the reconstruction stage with a dictionary-induced geometry, enabling our model to infer and learn the orthogonal vector **z** close to domain covariates. Since different directions in the latent space induced by the SAE dictionary may correspond to semantic factors with unequal scales and correlations, directly using the Euclidean inner product may lead to suboptimal similarity and orthogonality measurements that are inconsistent with the learned dictionary structure.

**Definition 1 (Latent Metric).** Suppose that the sparse autoencoder uses a linear decoder with tied weights, as specified in Eq. (4). Following standard quadratic forms induced by linear operators (Boyd & Vandenberghe, 2004), we define a dictionary-induced latent metric as

$$M = W_\theta^\top W_\theta. \qquad (5)$$

Here we measure similarity by the inner product $\langle a, b \rangle_M = a^\top M b$ and the norm $\|a\|_M^2 = a^\top M a$. Note that, $M$ in Eq. (5) is symmetric positive semi-definite, hence it defines a valid (pseudo-)inner product, see Appendix A.

This metric provides a notion consistent with the SAE reconstruction, and it will be used to define orthogonality and projection in the subsequent decomposition step.

**Objective 2: Reconstruction Loss.** Unlike standard $L_1$-norm SAEs, our goal is not only to learn a sparse code $s$ for interpretability, but also to induce a latent geometry through learned $W$ and $M$. This geometry will later serve as the basis for decomposing invariant and domain-specific components. In our model, we train the SAE by minimizing the reconstruction error measured under $M$, together with a sparsity regularization on the latent code $s$:

$$\mathcal{L}_{\mathrm{rec}} = \| v - \hat{v} \|_M^2 + \gamma \| s \|_1, \qquad (6)$$

where $\| v - \hat{v} \|_M^2 = (v - \hat{v})^\top M (v - \hat{v})$ evaluates the reconstruction discrepancy under the dictionary-induced geometry, and $\gamma$ controls the sparsity strength. By optimizing $\mathcal{L}_{\mathrm{rec}}$, the SAE learns a sparse reconstruction that facilitates interpretation, while empirically leading to a geometry that is consistent with the reconstruction directions.

To further justify the stability of the subsequent decomposition, we establish the following lemma, showing that a well-trained SAE makes the projection operation robust to reconstruction errors.

**Lemma 1 (Stability of $M$-Orthogonal Projection).** *Consider reconstructed $\hat{v}$ such that the reconstruction error $\|v - \hat{v}\|_M$ is bounded by a small constant $\delta > 0$, and suppose that $\|\hat{v}\|_M^2 + \varepsilon$ is bounded away from zero. With the induced metric $M$, the $M$-orthogonal projection of $v$ onto the direction spanned by $\hat{v}$ is given by $\Pi_{\hat{v}}^{(M)}(v) = \frac{\langle v, \hat{v} \rangle_M}{\|\hat{v}\|_M^2 + \varepsilon} \hat{v}$, where $\varepsilon > 0$ is a small stabilization constant. Then the induced projection is stable in the sense that*

$$\left\| \Pi_{\hat{v}}^{(M)}(v) - \Pi_v^{(M)}(v) \right\|_M \le C \| v - \hat{v} \|_M,$$

*where $C$ is a constant depending on $\|v\|_M$ and a lower bound of $\|\hat{v}\|_M^2 + \varepsilon$, but independent of $\|v - \hat{v}\|_M$.*

This lemma provides a sufficient condition under which using $\hat{v}$ as the reference direction yields a stable projection, which is desirable for our decomposition. The detailed proof is deferred to Appendix C.

### 3.4. Orthogonal Covariates Inference

After reconstructing $\hat{v}$, we go back and follow our motivation again to capture domain-specific variations in terms of $v$. Note that $v$ may still contain residual domain-dependent factors that are not eliminated by the label supervision. To model such residual variations on neural networks, we consider domain covariates $z$ as the orthogonal residual of $v$ with respect to $\hat{v}$. Thus, we define the domain-specific residual via an $M$-orthogonal projection.

**Definition 2 (Inference of Domain Covariates).** We initialize and compute the domain-covariant feature $z$ via an

$M$-orthogonal residual:

$$\hat{v} = W_\theta^\top s, \quad \alpha = \frac{\langle v, \hat{v}\rangle_M}{\|\hat{v}\|_M^2 + \varepsilon}, \quad z = v - \alpha\,\hat{v}, \quad (7)$$

where $\alpha$ measures the extent to which $v$ aligns with the reconstruction direction $\hat{v}$ under the dictionary-induced geometry, and $\varepsilon$ is a small constant for numerical stability.

With $z$ being $M$-orthogonal to $\hat{v}$, we apply another supervision on $z$ to encourage it to encode domain-specific factors, while $v$ remains optimized for label prediction. These distinct objectives promote a functional separation between invariant and covariant information without introducing additional neural modules. We provide operator properties and discuss possible degeneracy about $M$ in Appendix E and F, respectively. To further justify that Eq. (7) is not an ad-hoc construction, we show that $\alpha$ is the closed-form solution of an $M$-weighted projection with least squares, and $z$ is guaranteed to be $M$-orthogonal to $\hat{v}$.

**Proposition 1 (Closed-Form Solution).** *The coefficient $\alpha$ in Eq.* (7) *is the unique minimizer of a one-dimensional $M$-weighted least squares objective with $\varepsilon$-regularization:*

$$\alpha^* = \arg\min_\alpha \left( \|v - \alpha\,\hat{v}\|_M^2 + \varepsilon\,\alpha^2 \right) = \frac{\langle v, \hat{v}\rangle_M}{\|\hat{v}\|_M^2 + \varepsilon},$$

*where $\alpha^*$ represents the optimal solution and $z = v - \alpha^*\hat{v}$ satisfies $\langle z, \hat{v}\rangle_M = 0$ in the limit $\varepsilon \to 0$.*

The proof is provided in Appendix B. This proposition demonstrates that our residual design is close to an optimal projection. Such a property is crucial for our model, as it ensures that $z$ is orthogonal (under the $M$-metric) to $\hat{v}$. This geometric constraint encourages the domain classifier to rely on residual variations only. Next, we explicitly guide it to capture domain-specific information through a supervised domain classification objective.

**Objective 3: Domain Classification Loss.** We train a domain classifier $d_\omega$ on the residual feature $z$ with a binary domain indicator $\delta \in \{0,1\}$, where $\delta = 0$ denotes the source domain and $\delta = 1$ denotes the target domain:

$$\mathcal{L}_{\text{dcl}} = \mathbb{E}_{P_s}\big[\ell_{\text{CE}}(\omega; z, 0)\big] + \mathbb{E}_{P_t}\big[\ell_{\text{CE}}(\omega; z', 1)\big], \quad (8)$$

where $\ell_{\text{CE}}$ is the cross-entropy loss, and $z, z'$ are computed from $x, x'$ by Eq. (2) and Eq. (7), respectively.

Unlike prior disentanglement studies that suppress domain information by confusing a domain discriminator, we explicitly construct an $M$-orthogonal residual $z$ and directly supervise it to be domain-discriminative. Next, we supplement the following justification for why this procedure partitions domain-related information into $z$.

**Remark (Domain Information in $z$).** Assume the alignment objective yields a small distribution discrepancy in

the $v$-space, i.e., $\text{MMD}(P_s^v, P_t^v) \le \eta$, where $P_s^v$ and $P_t^v$ are the distributions of $v$ induced by $x \sim P_s$ and $x' \sim P_t$, respectively. Let $e_z$ denote the Bayes error of predicting the domain indicator $\delta$ from $z$. Then, under mild regularity conditions, the following relations provide an intuitive characterization of how domain information is partitioned by our residual inference:

$$I(\delta; z) \gtrsim 1 - h(e_z), \quad I(\delta; v) \lesssim C\,\eta^2,$$

where $h(\cdot)$ denotes the binary entropy and $C$ is a constant. Note that, $\gtrsim$ and $\lesssim$ are used to denote informal relations rather than strict or universally valid bounds.

Consequently, when alignment makes $\eta \to 0$ (so that $\delta$ becomes difficult to infer from $v$), while the domain classifier on $z$ becomes accurate (so that $e_z \to 0$), the domain-related information is *encouraged* to concentrate on $z$ rather than $v$. A more detailed discussion is deferred to Appendix D.

### 3.5. Training, Inference, and Interpretation

During training, we input a pair of data $x, x'$ from source and target domains and optimize the model in three following stages: (1) update $f_\phi$ and $p_\zeta$ by minimizing $\mathcal{L}_{\text{label}}$ only; (2) add $M$-induced SAE with $(W_\theta, h_\theta)$ by minimizing $\mathcal{L}_{\text{label}} + \lambda_2 \mathcal{L}_{\text{rec}}$; (3) update $f_\phi, p_\zeta$, and $d_\omega$ by minimizing $\mathcal{L}_{\text{label}} + \lambda_2 \mathcal{L}_{\text{rec}} + \lambda_3 \mathcal{L}_{\text{dcl}}$ only. We adopt their weighted linear combination as the training loss for each training stage. During inference, only input data $x'$ from target domain go through $f_\phi(\cdot)$ and $p_\zeta(\cdot)$ to predict $\hat{y}' = p_\zeta(f_\phi(x'))$. The pseudocode of `ExtraCare` can be found in Appendix H.

For model interpretation, ICD codes should be mapped to their respective meaningful dictionary features. Following the same setting in (Bricken et al., 2023), we get the ablated embedding $\tilde{s}^{(i,k)}$ by targeting top-$k$ activated dictionary features $s^{(i,k)} > 0$ for the $i$-th patient. We then recompute the ablated probabilities for all $\mathcal{C}$ classes, and compute the differences $\Delta\text{prob}^{(i,k)}(c)$, which helps us identify its most relevant ICD codes. More details are provided in Appendix J.1.

## 4. Experiments

In this section, we aim to address and answer the following three research questions (**RQs**):

**RQ1:** Can `ExtraCare` achieve robust and accurate predictions on target data under substantial domain shifts?

**RQ2:** To what extent does our method provide concept-grounded and human-understandable explanations that allow clinicians to know what information is transferred and what is domain-specific during adaptation?

**RQ3:** Can such explanations actually capture patient-level evidence and facility-specific shifts during deployment?

*Table 1.* Dataset Statistics (All Multi-Visit Patients).

| Item | eICU | OCHIN |
|---|---|---|
| # patients | 9,408 | 199,703 |
| Max. # visits | 7 | 51 |
| Avg. # visits | 2.15 | 13.42 |
| Max. # codes / visit | 54 | 31 |
| Avg. # codes / visit | 4.40 | 2.89 |
| Top5 frequent codes (ICD-9/10-CM) | 518.81, 584.9, 401.9, 486, 427.31 | I10, F41.9, F41.1, F17.200, F33.1 |
| # hospitals (facilities) | 187 | 2,936 |

## 4.1. Experimental Setup

**Clinical Predictive Tasks.** We evaluate our model on two clinical prediction tasks: diagnosis prediction and heart failure (HF) prediction. ① Diagnosis prediction is formulated as a multi-label classification task to predict medical codes for the next visit and is evaluated using weighted F1 score (w-$F_1$) and top-$k$ recall (R@k). ② HF prediction is treated as a binary classification problem and is evaluated using AUROC and F1 score due to imbalanced test data. Both tasks employ a sigmoid-activated prediction head trained with binary cross-entropy loss. Detailed task formulations and metric definitions are provided in Appendix I.1, and `ExtraCare` is implemented as described in Appendix I.2.

**Datasets.** We evaluate our model on two real-world EHR datasets: ① The eICU dataset (Pollard et al., 2018) contains multi-center ICU admissions from 208 hospitals across the United States between 2014 and 2015; ② the OCHIN dataset (DeVoe & Sears, 2013; DeVoe et al., 2014; Bensken et al., 2026) aggregates longitudinal EHR data from over 2,400 facilities across more than 40 states from 2012 to 2023. Table 1 presents basic statistics of the processed datasets. We extract 20,227 visits from 9,408 patients in eICU and 577,142 encounters from 199,703 patients in OCHIN. We elaborate on the cohort selection process and provide more dataset statistics in Appendix I.3.

**Baselines.** We evaluate our model with three kinds of baselines: (1) Inspired by domain generalization works (Wu et al., 2023; Hu et al., 2026a), the first category consists of naïve baselines, including **Oracle**, trained directly on the target data, and **Base**, trained solely on the source data. (2) The second category comprises feature adaptation methods. These include **DANN** (Ganin et al., 2016), **RMMD** (Wang et al., 2023), **RSDA** (Gu et al., 2024), and **BUA** (Chen et al., 2024), which focus on domain-shared features, **DAL** (Peng et al., 2019) and **RCG** (Liu et al., 2025), which rely on decomposition. **BUA** and **RCG** are recent works tackling clinical DA problems. (3) The third category consists of self-training methods, including **CST** (Liu et al., 2021) and **SSRT** (Sun et al., 2022a), which leverage pseudo labels on the target domain to iteratively refine decision boundary. Note that, diagnosis prediction is not a multi-class classification, and we thus drop RCG and CST as they do not fit into the setting. More details of baselines and their implementation in our setting can be found in Appendix I.5.

## 4.2. Predictive Robustness in Adaptation (RQ1)

Prior works assume that distributional shifts are primarily induced by spatial and temporal gaps (Wu et al., 2023). Concretely, spatial shifts are attributed to cross-institute changes shaped by regional healthcare systems, while temporal shifts capture changes in patient populations over time. Following this common setup, we evaluate our model under these two shifts (see Appendix I.4). For spatial evaluation on the eICU dataset, hospitals in the West or Midwest are treated as the target test data, while the remaining groups (Northeast and South) serve as the source training data. For temporal adaptation, we use the OCHIN dataset, as eICU only spans two years (2014-2015): patients admitted after 2021 are used as the target test data, while all preceding patients (2012-2021) are included in the source training data.

Table 2 summarizes the domain adaptation results on the eICU and OCHIN datasets. First, a substantial performance gap is observed between Oracle and Base, indicating the presence of spatial and temporal domain gaps. This supports the use of the DA setting. Second, feature adaptation baselines improve over Base by learning domain-shared representations, and we note that DANN and DAL with the simplest architecture achieve minimal improvements. Third, the two self-training baselines, CST and SSRT, also outperform Base, but the improvement is less stable than results of other methods. This outcome is expected because self-training adapts models in multiple steps during training. Lastly, `ExtraCare` outperforms all baselines for almost all tasks. Specifically, in terms of the F1 scores, our method achieves a 12% relative gain in eICU diagnosis prediction, 9% in eICU HF prediction, 16% in OCHIN diagnosis prediction, and 14% in OCHIN HF prediction. We also extend our evaluation under spatial shifts on the OCHIN dataset (see Table 8 in Appendix I.6).

## 4.3. Model Interpretation (RQ2)

Figure 2 illustrates how we map sparse latent dimensions to clinically meaningful conditions and diagnose domain sensitivity. First, we randomly select two patients with their sparse vectors, then choose the top-3 dimensions with the highest activations for ablation, as shown in Figure 2(a). Then, we compute the probability differences in diagnoses and domain partition $\Delta$prob for each selected dimension (see details in Appendix J.1). Figure 2(b) shows that a dimension typically affects only several ICD codes above $\Delta$prob $= 0.05$ (dashed line), while most mapped codes remain weakly sensitive, forming concise and clinically coherent attributions. For example, the activation on the 237-th dimension mostly contributes to `F41.1` (Generalized Anxi-

*Table 2.* **Results of domain adaptation on the eICU and OCHIN datasets.** We report the average performance (%) and the standard error (in bracket). The results show that `ExtraCare` exhibits robustness against spatial (eICU) and temporal (OCHIN) domain shifts. Moreover, we use **bold** font to indicate the best model and use highlighted gray color to denote the second-best method.

| Model | eICU | | | | OCHIN | | | |
| | Diagnosis | | Heart Failure | | Diagnosis | | Heart Failure | |
| | w-$F_1$ | R@10 | AUROC | F1 | w-$F_1$ | R@5 | AUROC | F1 |
|---|---|---|---|---|---|---|---|---|
| Oracle (Upper Bound) | 69.72 (0.10) | 84.66 (0.05) | 92.54 (0.20) | 80.17 (0.03) | 76.14 (0.17) | 83.78 (0.12) | 97.22 (0.03) | 86.52 (0.16) |
| Base (Lower Bound) | 61.09 (0.02) | 76.54 (0.14) | 84.52 (0.03) | 72.76 (0.04) | 63.77 (0.13) | 75.61 (0.25) | 91.40 (0.04) | 74.88 (0.09) |
| DANN (Ganin et al., 2016) | 62.14 (0.19) | 77.32 (0.29) | 85.43 (0.18) | 74.29 (0.13) | 66.81 (0.30) | 76.32 (0.02) | 92.81 (0.26) | 76.06 (0.09) |
| DAL (Peng et al., 2019) | 61.58 (0.05) | 76.74 (0.04) | 87.02 (0.09) | 73.92 (0.25) | 65.42 (0.06) | 76.49 (0.07) | 93.48 (0.19) | 77.44 (0.18) |
| CST (Liu et al., 2021) | / | / | 87.34 (0.14) | 74.81 (0.21) | / | / | 92.23 (0.19) | 78.06 (0.10) |
| SSRT (Sun et al., 2022a) | 62.04 (0.14) | 77.96 (0.23) | 86.22 (0.17) | 75.83 (0.11) | 71.13 (0.34) | 80.93 (0.29) | 94.56 (0.22) | 83.04 (0.25) |
| RMMD (Wang et al., 2023) | 64.34 (0.18) | 79.07 (0.14) | 86.51 (0.29) | 76.13 (0.02) | 68.48 (0.26) | 78.65 (0.20) | 92.06 (0.03) | 80.90 (0.26) |
| RSDA (Gu et al., 2024) | 62.96 (0.06) | 78.25 (0.21) | 85.91 (0.20) | 74.64 (0.03) | 70.11 (0.08) | 80.50 (0.05) | 94.78 (0.12) | 82.94 (0.25) |
| BUA (Chen et al., 2024) | 63.57 (0.12) | 78.63 (0.27) | 89.77 (0.06) | **79.83 (0.12)** | 71.77 (0.15) | 81.04 (0.23) | 95.05 (0.28) | 78.13 (0.04) |
| RCG (Liu et al., 2025) | / | / | 88.26 (0.10) | 77.51 (0.13) | / | / | 93.55 (0.25) | 81.85 (0.18) |
| `ExtraCare` | **68.61 (0.11)** | **82.19 (0.08)** | 91.88 (0.08) | 79.64 (0.07) | **74.05 (0.06)** | **82.89 (0.12)** | 95.48 (0.17) | 85.38 (0.15) |
| - Ours (w/o $\mathcal{L}_2$ & $\mathcal{L}_3$) | 64.87 (0.26) | 79.63 (0.17) | 87.18 (0.29) | 76.74 (0.21) | 69.26 (0.23) | 79.17 (0.28) | 93.07 (0.24) | 81.36 (0.19) |
| - Ours (w/o $W, z$ & $\mathcal{L}_3$) | 66.73 (0.18) | 80.96 (0.22) | 89.64 (0.15) | 78.46 (0.27) | 71.94 (0.25) | 81.13 (0.16) | 94.43 (0.21) | 83.92 (0.30) |
| - Ours (w/o $M, z_{\perp p}$) | 67.58 (0.12) | 81.47 (0.24) | 90.69 (0.20) | 79.08 (0.18) | 73.02 (0.29) | 82.06 (0.14) | 95.07 (0.19) | 84.72 (0.22) |
| - Ours (w/o $\mathcal{L}_3$) | 66.14 (0.30) | 80.54 (0.19) | 88.92 (0.25) | 77.93 (0.16) | 71.28 (0.20) | 80.76 (0.27) | 94.04 (0.26) | 83.34 (0.17) |

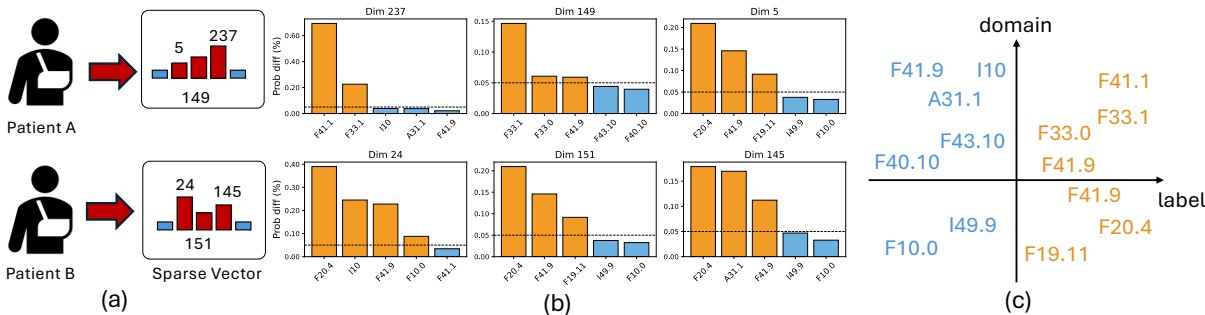

*Figure 2.* **Clinical Concept Attribution via Sparse-Dimension Ablation.** (a) We extract sparse concept activations for two patients and select the top-3 active (with highest activations) dimensions. (b) We ablate each selected dimension and visualize the resulting diagnosis absolute probability change $\Delta$prob, with a threshold of 0.05 (dashed line). (c) We categorize mapped ICD10-CM codes by label impact and domain sensitivity to distinguish transferable evidence from shift-sensitive variation.

ety Disorder) and `F33.1` (Depressive Disorder). We also characterize domain sensitivity (whether $\Delta$prob $> 0.05$) to understand which codes are likely to reflect transferable predictive evidence versus domain-specific variation. Figure 2(c) organizes the mapped codes into a four-quadrant diagram for the patient A, yielding four meaningful cases:

1) **High label impact & Low domain impact:** These codes are strongly driven by the sparse dimension while remaining relatively stable across domains, making them reliable for cross-domain generalization.

2) **High label impact & High domain impact:** These codes substantially influence prediction, yet their patterns vary across domains, suggesting cohort-specific effects that may compromise robustness if not explicitly handled.

3) **Low label impact & High domain impact:** These codes are not key drivers of predicted outcomes, but they exhibit strong domain sensitivity, making them useful signals for detecting domain discrepancy.

4) **Low label impact & Low domain impact:** These codes show limited contribution to prediction and adaptation.

Note that, the same ICD10-CM code may appear in different quadrants. For instance, `F41.9` (Unspecified Anxiety Disorder) appears in both domain-sensitive and label-sensitive dimensions in Figure 2(c). This suggests that anxiety-related conditions can serve as robust predictive evidence in some latent clinical concepts, while also acting as shift-sensitive signals in others. We extend our discussion about qualitative analysis and clinical interpretation from Appendix J.2 to J.4.

### 4.4. External Grounding of Explanations (RQ3)

To test whether the explanations are grounded beyond the model's internal scoring convention, we evaluate them in the facility adaptation setting on the OCHIN dataset, where the source cohort is Primary Care/Family Practice and the targets are specialized facilities. We compare `ExtraCare` with SHAP (Lundberg & Lee, 2017), Integrated Gradients

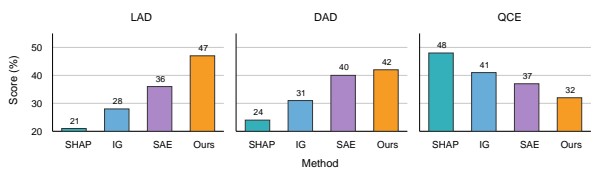

*Figure 3.* **Quantitative external grounding of adaptation explanations.** LAD and DAD measure whether the label and domain axes align with patient relevance and facility specificity, respectively. QCE measures distance from ideal audit-group anchors, where lower is better (see details in Appendix J.5).

*Table 3.* Preference-based validation of four-quadrant explanations against regular SAE. QRS denotes quadrant rule satisfaction.

| Judge | QRS/4 | Coherence | Partition | Preferred (%) |
|---|---|---|---|---|
| LLM (SAE) | 2.48 | 3.56 | 3.24 | 32 |
| LLM (ExtraCare) | **3.64** | **4.24** | **4.08** | **68** |
| Human (SAE) | 2.36 | 3.44 | 3.16 | 36 |
| Human (ExtraCare) | **3.32** | **4.08** | **3.92** | **64** |

(IG) (Sundararajan et al., 2017), and a regular SAE baseline (Huben et al., 2024). For each patient, all methods produce label-related and domain-related code scores. We convert these scores into a controlled two-axis audit view by assigning candidate ICD-10 codes to high/low label-evidence groups and high/low domain-sensitivity groups, yielding four audit groups: high-label/high-domain, high-label/low-domain, low-label/high-domain, and low-label/low-domain. Since SHAP and IG produce dense attributions while SAE-based methods expose sparse code sets, we use the same per-patient candidate-code budget before forming these groups.

We define two external anchors. The first is patient relevance, $R_i(c) = \mathbb{I}[c \in Y_i \cup H_i]$, where $Y_i$ and $H_i$ denote the future and historical diagnosis sets of patient $i$. The second is facility specificity, which measures whether code $c$ is enriched in target facility $t$ relative to the source cohort. Intuitively, high-label groups should contain more patient-relevant codes, while high-domain groups should contain more facility-specific codes. We therefore report Label-Axis Difference (LAD), Domain-Axis Difference (DAD), and Quadrant Calibration Error (QCE). LAD and DAD evaluate separation along the two axes, while QCE evaluates how close the four audit-group centroids are to their intended anchor locations. Additional implementation details are provided in Appendix J.5.

Figure 3 shows that ExtraCare produces the strongest external grounding. Compared with SAE, ExtraCare improves LAD from 36% to 47%, slightly improves DAD from 40% to 42%, and reduces QCE from 37% to 32%. The gap is larger against SHAP and IG, suggesting that dense attribution methods can identify relevant codes but do not separate patient relevance and facility specificity as cleanly. We also evaluate whether the explanations are easier to

*Table 4.* Cosine similarity of class and domain classifiers trained on encoded features $v_0$, $v$, and $z$ on OCHIN (average over all classes). In this case, $v_0$ represents the full encoded feature without any adaptation techniques (i.e., optimized by the supervised loss only).

| Cosine Similarity | Diagnosis Prediction | | HF Prediction | |
|---|---|---|---|---|
| | Source | Target | Source | Target |
| $W_c(v_0)$ v.s. $W_d(v_0)$ | 0.0496 | / | 0.0383 | / |
| $W_c(v_0)$ v.s. $W_c(z)$ | 0.0742 | 0.0973 | 0.0681 | 0.0719 |
| $W_c(v_0)$ v.s. $W_c(v)$ | 0.6115 | 0.4914 | 0.7526 | 0.6773 |

judge as coherent clinical summaries by comparing regular SAE and ExtraCare in anonymized paired cases. LLM judges evaluate 60 patients per target facility (300 total), and two human evaluators with medical training evaluate 5 patients per target facility (25 total). As shown in Table 3, ExtraCare obtains higher Quadrant Rule Satisfaction, clinical coherence, and partition clarity, and is preferred in 68% of LLM judgments and 64% of human judgments. Together, these results support the use of a shared sparse concept basis with explicit label-domain factorization.

### 4.5. Ablation Study

We conduct ablation study for ExtraCare by removing or replacing key modules, and also attach the results in Table 2. ① w/o $\mathcal{L}_{rec}$ & $\mathcal{L}_{dcl}$: retains the label supervision with the alignment regularization only, and the performance drop indicates that both modules are essential for adaptation; ② w/o $W, z$ & $\mathcal{L}_{dcl}$: disables $W, z$ and $\mathcal{L}_{dcl}$ for orthogonal inference, and the worse results show that learning invariant features only is not enough for model generalization; ③ w/o $M, z_{\perp p}$: replaces $M$ with a Euclidean alternative ($z_{\perp p}$ is no longer enforced), and a moderate degradation suggests that the orthogonal constraints help model distinguish task-relevant and domain-related factors; ④ w/o $\mathcal{L}_{dcl}$: defines $\mathcal{L}_{dcl}$ with an adversarial cross-entropy loss on y, and the worse results imply that adaptation benefits from learning invariant features and covariates under distinct objectives. In Appendix K.1, we further intervene on the residual component $z$ at inference time and show that masking it consistently degrades target performance under facility shifts. Appendix K.2 also verifies that the interpretation based on $\Delta$prob is stable around the threshold used in Figure 2.

### 4.6. Effectiveness of Factorized Subspaces

Following the setting in (Shen et al., 2022), we conduct a linear-probe analysis on encoded features: we first train linear classifiers on pre-trained embeddings ($v_0$, $v$, and $z$) to predict either classes or domain IDs, and then report the cosine similarity between learned classifier weights $W_{c,d}(\cdot)$. The subscripts $c$ and $d$ denote the class classifier and the domain classifier, respectively. For stable probing and bal-

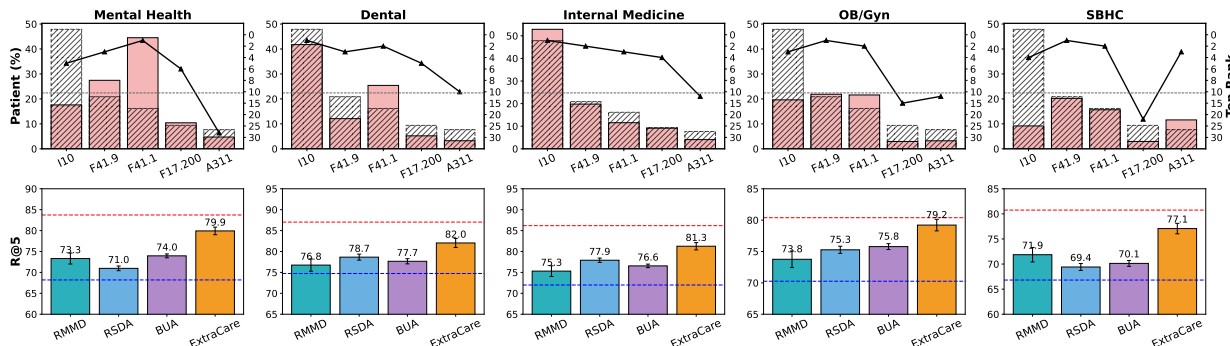

*Figure 4.* **Facility-specific ICD-10 code distribution shift and cross-domain diagnosis retrieval performance.** The top row visualizes the Top 5 frequent ICD10 codes from the source (Primary Care/Family Practice) subsets, showing patient proportions (bars) and rank frequency (line) across facility-specific cohorts, with hatched bars indicating the source-subset frequencies for comparison. The bottom row reports R@5 across different models when adapting models on specific type, where blue and red lines denote Base and Oracle.

anced sample sizes, we randomly sample 10,000 patients from both source and target subset. In Table 4, the first row displays the same pattern as in previous findings: linear classifiers trained to predict the class are nearly orthogonal to the linear weights learned by domain classifiers. Here we merge those two subsets when training the domain classifier, so the similarity entry is only reported for one domain. The second row also has very low similarity, indicating that covariates $z$ carry limited label information. The third row shows much higher similarity with $v$, implying that $v$ preserves the label information while improving alignment across domains (reflected by comparable similarity in source and target columns). Note that the cosine similarity values from different tasks are not comparable.

### 4.7. Adaptation From General to Specific Facilities

In clinical practices, model adaptation is rarely driven by either spatial or temporal shifts, but more often by changes from primary care populations to specialized facility cohorts. This motivates us to extend an evaluation on OCHIN across different facility types. Concretely, we merge records in *Primary Care* and *Family Practice* institutes and randomly sample 30,000 patients as the source domain, while treating *Mental Health*, *Dental*, *Internal Medicine*, *OB/Gyn* (Obstetrics & Gynecology), and *SBHC* (School-Based Health Center) as target domains. The results of diagnosis prediction are shown in Figure 4. The top row reveals label distribution across facilities, as the Top-5 frequent ICD-10 codes from the source cohort exhibit different prevalence and rank patterns in each target cohort. Compared to the settings in Table 2, facility-based partition induces more challenging distribution shifts, as different facilities tend to treat distinct patient populations with different disease profiles. Hence, all models suffer performance drops under these shifts (on the bottom row), with the largest degradation observed in Mental Health and SBHC, consistent with their stronger distribution divergence. Nevertheless,

`ExtraCare` still consistently outperforms other DA baselines and shows better resilience against degradation.

## 5. Conclusion

In this work, we propose `ExtraCare`, a transparent domain adaptation framework that decomposes patient representations into label and domain information upon sparse vectors and residual inference, enabling both robust transfer and concept-grounded interpretability. On real-world EHR datasets, eICU and OCHIN, our method consistently demonstrates superiority over other DA baselines on two diagnostic tasks under spatial and temporal distribution gaps. We also extend a more practical setting for evaluation across facilities, where `ExtraCare` maintains stronger robustness and exhibits smaller degradation. We further validate the interpretation pipeline under facility adaptation with patient-relevance and facility-specificity anchors, showing stronger axis separation and clearer judged explanations than standard XAI baselines. Overall, the proposed method shows robust adaptation while offering prevalence-aware interpretation, where dictionary features jointly reveal transferable invariants and domain covariates from patient cases.

## Impact Statement

This paper presents work whose goal is to advance the field of domain adaptation in clinical prediction. There are many potential societal consequences of our work, none of which we feel must be specifically highlighted here.

## Acknowledgment

This research was, in part, funded by the US National Institutes of Health (NIH) Agreement No. 1OT2OD032581-01 and National Science Foundation (NSF) under grants 2047843 and 2437621. The views and conclusions contained in this document are those of the authors and should

not be interpreted as representing the official policies, either expressed or implied, of the NIH and NSF. The research reported in this work was powered by PCORnet®. PCORnet has been developed with funding from the Patient-Centered Outcomes Research Institute® (PCORI®) and conducted with the Accelerating Data Value Across a National Community Health Center Network (ADVANCE) Clinical Research Network (CRN). ADVANCE is a Clinical Research Network in PCORnet® led by OCHIN in partnership with Health Choice Network, Fenway Health, University of Washington, and Oregon Health & Science University. ADVANCE's participation in PCORnet® is funded through the PCORI Award RI-OCHIN-01-MC.

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

## A. Basic Properties of the Dictionary-Induced Metric

In this section, we justify that the SAE-induced matrix

$$M := W_\theta^\top W_\theta$$

defines a valid (pseudo-)inner product and (semi-)norm used throughout the orthogonal residual inference.

**Symmetry.** By construction,

$$M^\top = (W_\theta^\top W_\theta)^\top = W_\theta^\top W_\theta = M,$$

hence $M$ is symmetric.

**Positive semidefiniteness and the induced seminorm.** For any vector $a \in \mathbb{R}^d$,

$$a^\top M a = a^\top W_\theta^\top W_\theta a = (W_\theta a)^\top (W_\theta a) = \|W_\theta a\|_2^2 \geq 0.$$

Therefore, $M \succeq 0$ and $\langle a, b \rangle_M := a^\top M b$ defines a valid *bilinear form*. Moreover, $\|a\|_M := \sqrt{a^\top M a}$ defines a *seminorm* in general.

**When is it a proper inner product / norm?** The seminorm becomes a proper norm (equivalently, $\langle \cdot, \cdot \rangle_M$ becomes an inner product) if and only if $M$ is positive definite, i.e.,

$$a^\top M a = 0 \Rightarrow a = 0.$$

Since $a^\top M a = \|W_\theta a\|_2^2$, we have

$$a^\top M a = 0 \Leftrightarrow W_\theta a = 0 \Leftrightarrow a \in \text{Null}(W_\theta).$$

Hence, $M \succ 0$ holds if and only if $\text{Null}(W_\theta) = \{0\}$, i.e., $W_\theta$ has full column rank. When $W_\theta$ is rank-deficient, $M$ is only positive semidefinite and the induced geometry is degenerate on $\text{Null}(W_\theta)$.

**Role of the $\varepsilon$-stabilization in projection.** Our residual inference uses the coefficient

$$\alpha = \frac{\langle v, \hat{v} \rangle_M}{\|\hat{v}\|_M^2 + \varepsilon}, \qquad \varepsilon > 0,$$

which is well-defined regardless of whether $M$ is singular. In particular, the additive $\varepsilon$ guarantees a strictly positive denominator even when $\|\hat{v}\|_M^2$ is small (e.g., early in training or under degenerate directions), and can be interpreted as a standard Tikhonov-style stabilization term. This stabilization will also appear explicitly in the projection stability bound proved in the subsequent section.

## B. Proof of Proposition 1 (Closed-Form $M$-Projection)

We consider the residual construction in Eq. (7):

$$\alpha = \frac{\langle v, \hat{v} \rangle_M}{\|\hat{v}\|_M^2 + \varepsilon}, \qquad z = v - \alpha \hat{v},$$

where $\langle a, b \rangle_M := a^\top M b$ and $\|a\|_M^2 := a^\top M a$.

We first show that $\alpha$ is the unique minimizer of a one-dimensional $\varepsilon$-regularized weighted least-squares problem:

$$\alpha^\star = \arg\min_{\alpha \in \mathbb{R}} \|v - \alpha \hat{v}\|_M^2 + \varepsilon \alpha^2.$$

Expanding the objective yields

$$\begin{aligned}
\|v - \alpha \hat{v}\|_M^2 + \varepsilon \alpha^2 &= (v - \alpha \hat{v})^\top M (v - \alpha \hat{v}) + \varepsilon \alpha^2 \\
&= v^\top M v - 2\alpha\, v^\top M \hat{v} + \alpha^2 \hat{v}^\top M \hat{v} + \varepsilon \alpha^2 \\
&= \text{const} - 2\alpha \langle v, \hat{v} \rangle_M + \alpha^2 (\|\hat{v}\|_M^2 + \varepsilon).
\end{aligned}$$

Since $\|\hat{v}\|_M^2 + \varepsilon > 0$, this is a strictly convex quadratic in $\alpha$, so the minimizer is unique. Setting the derivative to zero gives

$$-2\langle v, \hat{v}\rangle_M + 2\alpha(\|\hat{v}\|_M^2 + \varepsilon) = 0 \quad \Rightarrow \quad \alpha^\star = \frac{\langle v, \hat{v}\rangle_M}{\|\hat{v}\|_M^2 + \varepsilon},$$

which matches Eq. (7).

Next, we establish the orthogonality property of the residual. Let $z = v - \alpha^\star \hat{v}$. Then

$$\langle z, \hat{v}\rangle_M = \langle v - \alpha^\star \hat{v}, \hat{v}\rangle_M = \langle v, \hat{v}\rangle_M - \alpha^\star \|\hat{v}\|_M^2.$$

If $\varepsilon = 0$, substituting $\alpha^\star = \langle v, \hat{v}\rangle_M / \|\hat{v}\|_M^2$ yields $\langle z, \hat{v}\rangle_M = 0$, i.e., exact $M$-orthogonality. If $\varepsilon > 0$, we obtain

$$\langle z, \hat{v}\rangle_M = \langle v, \hat{v}\rangle_M \left(1 - \frac{\|\hat{v}\|_M^2}{\|\hat{v}\|_M^2 + \varepsilon}\right) = \langle v, \hat{v}\rangle_M \cdot \frac{\varepsilon}{\|\hat{v}\|_M^2 + \varepsilon},$$

showing that the deviation from exact orthogonality is controlled by $\varepsilon$ and vanishes as $\varepsilon \to 0$. This completes the proof.

## C. Proof of Lemma 1 (Stability of $M$-Orthogonal Projection)

Recall that for any nonzero direction $u$, we define the (regularized) $M$-projection coefficient of $v$ onto $u$ as

$$\alpha(u; v) := \frac{\langle v, u\rangle_M}{\|u\|_M^2 + \varepsilon}, \qquad \varepsilon > 0,$$

and the corresponding projected component as

$$\Pi_u^{(M)}(v) := \alpha(u; v)\, u.$$

Lemma 1 states that the projection operator is stable with respect to perturbations of the reference direction, i.e., there exists a constant $C > 0$ such that

$$\left\|\Pi_{\hat{v}}^{(M)}(v) - \Pi_v^{(M)}(v)\right\|_M \leq C \|v - \hat{v}\|_M,$$

where $C$ depends on $\|v\|_M$ and the lower bound of $\|\hat{v}\|_M^2 + \varepsilon$.

We begin by expanding the difference:

$$\Pi_{\hat{v}}^{(M)}(v) - \Pi_v^{(M)}(v) = \alpha(\hat{v}; v)\, \hat{v} - \alpha(v; v)\, v.$$

Add and subtract $\alpha(\hat{v}; v)\, v$ to obtain

$$\alpha(\hat{v}; v)\, \hat{v} - \alpha(v; v)\, v = \alpha(\hat{v}; v)\, (\hat{v} - v) + \big(\alpha(\hat{v}; v) - \alpha(v; v)\big)\, v.$$

Taking the $M$-norm and applying the triangle inequality yields

$$\left\|\Pi_{\hat{v}}^{(M)}(v) - \Pi_v^{(M)}(v)\right\|_M \leq |\alpha(\hat{v}; v)|\, \|\hat{v} - v\|_M + |\alpha(\hat{v}; v) - \alpha(v; v)|\, \|v\|_M. \tag{9}$$

We now bound the two terms on the right-hand side.

**Bounding $|\alpha(\hat{v}; v)|$.** By the Cauchy–Schwarz inequality under the $M$-inner product,

$$|\langle v, \hat{v}\rangle_M| \leq \|v\|_M \|\hat{v}\|_M.$$

Therefore,

$$|\alpha(\hat{v}; v)| = \frac{|\langle v, \hat{v}\rangle_M|}{\|\hat{v}\|_M^2 + \varepsilon} \leq \frac{\|v\|_M \|\hat{v}\|_M}{\|\hat{v}\|_M^2 + \varepsilon} \leq \frac{\|v\|_M}{\sqrt{\varepsilon}}, \tag{10}$$

where the last inequality uses $\|\hat{v}\|_M^2 + \varepsilon \geq \varepsilon$.

**Bounding $|\alpha(\hat{v}; v) - \alpha(v; v)|$.** We compute

$$\alpha(\hat{v}; v) - \alpha(v; v) = \frac{\langle v, \hat{v} \rangle_M}{\|\hat{v}\|_M^2 + \varepsilon} - \frac{\langle v, v \rangle_M}{\|v\|_M^2 + \varepsilon}.$$

Add and subtract $\frac{\langle v, v \rangle_M}{\|\hat{v}\|_M^2 + \varepsilon}$:

$$\alpha(\hat{v}; v) - \alpha(v; v) = \frac{\langle v, \hat{v} - v \rangle_M}{\|\hat{v}\|_M^2 + \varepsilon} + \langle v, v \rangle_M \left( \frac{1}{\|\hat{v}\|_M^2 + \varepsilon} - \frac{1}{\|v\|_M^2 + \varepsilon} \right).$$

For the first term, again by Cauchy–Schwarz,

$$\left| \frac{\langle v, \hat{v} - v \rangle_M}{\|\hat{v}\|_M^2 + \varepsilon} \right| \leq \frac{\|v\|_M \|\hat{v} - v\|_M}{\|\hat{v}\|_M^2 + \varepsilon}. \tag{11}$$

For the second term, note that

$$\left| \frac{1}{\|\hat{v}\|_M^2 + \varepsilon} - \frac{1}{\|v\|_M^2 + \varepsilon} \right| = \frac{\left| \|v\|_M^2 - \|\hat{v}\|_M^2 \right|}{(\|\hat{v}\|_M^2 + \varepsilon)(\|v\|_M^2 + \varepsilon)}.$$

Moreover,

$$\left| \|v\|_M^2 - \|\hat{v}\|_M^2 \right| = \left| \langle v, v \rangle_M - \langle \hat{v}, \hat{v} \rangle_M \right| = \left| \langle v - \hat{v}, v + \hat{v} \rangle_M \right| \leq \|v - \hat{v}\|_M \|v + \hat{v}\|_M.$$

Thus, using $\langle v, v \rangle_M = \|v\|_M^2$,

$$\left| \langle v, v \rangle_M \left( \frac{1}{\|\hat{v}\|_M^2 + \varepsilon} - \frac{1}{\|v\|_M^2 + \varepsilon} \right) \right| \leq \frac{\|v\|_M^2 \|v - \hat{v}\|_M \|v + \hat{v}\|_M}{(\|\hat{v}\|_M^2 + \varepsilon)(\|v\|_M^2 + \varepsilon)}. \tag{12}$$

Combining (11)–(12) gives

$$|\alpha(\hat{v}; v) - \alpha(v; v)| \leq \frac{\|v\|_M \|v - \hat{v}\|_M}{\|\hat{v}\|_M^2 + \varepsilon} + \frac{\|v\|_M^2 \|v + \hat{v}\|_M}{(\|\hat{v}\|_M^2 + \varepsilon)(\|v\|_M^2 + \varepsilon)} \|v - \hat{v}\|_M. \tag{13}$$

**Final bound.** Substituting (10) and (13) into (9), we obtain

$$\left\| \Pi_{\hat{v}}^{(M)}(v) - \Pi_v^{(M)}(v) \right\|_M \leq \left( \frac{\|v\|_M}{\sqrt{\varepsilon}} + \|v\|_M \left[ \frac{\|v\|_M}{\|\hat{v}\|_M^2 + \varepsilon} + \frac{\|v\|_M^2 \|v + \hat{v}\|_M}{(\|\hat{v}\|_M^2 + \varepsilon)(\|v\|_M^2 + \varepsilon)} \right] \right) \|v - \hat{v}\|_M.$$

This proves the stated stability inequality with a constant $C$ depending on $\|v\|_M$ and the denominator lower bounds $\|\hat{v}\|_M^2 + \varepsilon$ and $\|v\|_M^2 + \varepsilon$. In particular, the regularization $\varepsilon > 0$ ensures the bound remains finite even when $\|\hat{v}\|_M$ is small, and the projection mapping is Lipschitz in the reference direction. $\qquad\square$

## D. Further Discussion for Remark (Information-Theoretic Intuition)

This section provides additional intuition supporting Remark 2 in the main text. We emphasize that the relations below are not intended as universally tight inequalities without additional assumptions; rather, they serve to connect (i) domain predictability from $z$ and (ii) alignment in the $v$-space to the qualitative claim that domain-related information is encouraged to concentrate in $z$ rather than $v$.

**From Bayes domain error on $z$ to a lower bound on $I(\delta; z)$.** Assume $\delta \in \{0, 1\}$ is a binary domain indicator and consider predicting $\delta$ from $z$. Let $e_z$ denote the Bayes error of the optimal domain classifier given $z$. A classical relationship between classification error and conditional entropy is given by Fano-type inequalities (Hellman & Raviv, 1970; Feder & Merhav, 1994), which (for the binary case) imply that smaller Bayes error leads to smaller conditional entropy $H(\delta \mid z)$. Since mutual information satisfies

$$I(\delta; z) = H(\delta) - H(\delta \mid z),$$

when the domain classifier becomes accurate (i.e., $e_z \to 0$), the uncertainty $H(\delta \mid z)$ decreases, and hence $I(\delta; z)$ increases. When the two domains are approximately balanced so that $H(\delta) \approx 1$ bit, this yields the intuition that

$$I(\delta; z) \text{ approaches } 1 \text{ as } e_z \to 0,$$

consistent with the first part of Remark 2.

**From alignment in $v$-space to reduced domain information in $v$.** The alignment objective enforces a small discrepancy between the induced feature distributions across domains, e.g.,

$$\text{MMD}(P_s^v, P_t^v) \leq \eta.$$

Intuitively, if $P_s^v$ and $P_t^v$ become indistinguishable, then it becomes difficult to infer the domain indicator $\delta$ from $v$, which suggests that the mutual information $I(\delta; v)$ should be small. Such a connection can be formalized under additional regularity conditions that relate distributional discrepancy metrics (including MMD with characteristic kernels) to hypothesis-test or classification distinguishability between $P_s^v$ and $P_t^v$ (Ben-David et al., 2010; Gretton et al., 2012). In particular, when $\eta \to 0$, the optimal domain prediction error given $v$ tends to increase toward chance level, which implies $H(\delta \mid v)$ increases and therefore $I(\delta; v)$ decreases.

**Why domain information is encouraged to concentrate in $z$.** In our design, $z$ is defined as an $M$-orthogonal residual component with respect to the reconstruction direction $\hat{v}$, and we explicitly train a domain classifier on $z$. Thus, the model is incentivized to place domain-discriminative information into the residual component $z$, while simultaneously reducing the domain discrepancy in the aligned feature space $v$. Together, these effects provide a geometric and information-theoretic intuition that domain-related variability is encouraged to concentrate in $z$ rather than $v$, consistent with our Remark.

## E. Additional Properties of the $M$-Projection Operator

In this section, we provide several useful properties of the $M$-projection operator used in our residual inference. These properties are standard for weighted least-squares projections and help clarify the geometric role of the coefficient in Eq. (7).

Recall the $\varepsilon$-regularized coefficient

$$\alpha(u; v) := \frac{\langle v, u \rangle_M}{\|u\|_M^2 + \varepsilon}, \qquad \varepsilon > 0,$$

and the corresponding projected component

$$\Pi_u^{(M)}(v) := \alpha(u; v)\, u, \qquad r_u^{(M)}(v) := v - \Pi_u^{(M)}(v),$$

where $r_u^{(M)}(v)$ is the residual.

**(1) Weighted least-squares optimality.** For any fixed nonzero $u$, $\alpha(u; v)$ is the unique minimizer of the one-dimensional quadratic problem

$$\alpha(u; v) = \arg\min_{\alpha \in \mathbb{R}} \|v - \alpha u\|_M^2 + \varepsilon \alpha^2.$$

Equivalently, $\Pi_u^{(M)}(v)$ is the best $\varepsilon$-regularized approximation to $v$ in the subspace $\text{span}\{u\}$ under the $M$-geometry.

**(2) Approximate orthogonality of the residual.** Let $r = r_u^{(M)}(v) = v - \alpha(u; v)u$. Then

$$\langle r, u \rangle_M = \langle v, u \rangle_M - \alpha(u; v)\|u\|_M^2 = \langle v, u \rangle_M \cdot \frac{\varepsilon}{\|u\|_M^2 + \varepsilon}.$$

Hence, for $\varepsilon = 0$ we recover exact $M$-orthogonality $\langle r, u \rangle_M = 0$, and for $\varepsilon > 0$ the deviation is controlled by $\varepsilon$ and vanishes as $\varepsilon \to 0$.

**(3) Linearity in $v$.** For any fixed $u$, the mapping $v \mapsto \Pi_u^{(M)}(v)$ is linear. Indeed,

$$\Pi_u^{(M)}(v) = \frac{\langle v, u \rangle_M}{\|u\|_M^2 + \varepsilon}\, u = \frac{u\, u^\top M}{\|u\|_M^2 + \varepsilon}\, v,$$

which is an affine-free linear operator on $v$.

**(4) Operator form and rank-one structure.** Define the rank-one matrix

$$P_u^{(M)} := \frac{u\, u^\top M}{\|u\|_M^2 + \varepsilon}.$$

Then $\Pi_u^{(M)}(v) = P_u^{(M)}v$ and the residual mapping is $(I - P_u^{(M)})v$. This representation makes it explicit that the projection extracts the component of $v$ aligned with $u$ under the $M$-inner product.

**(5) Limit as $\varepsilon \to 0$.** When $\varepsilon \to 0$ and $\|u\|_M^2 > 0$, the operator reduces to the standard $M$-orthogonal projection onto $\text{span}\{u\}$:

$$\Pi_u^{(M)}(v) \to \frac{\langle v, u \rangle_M}{\|u\|_M^2}\, u, \qquad \langle v - \Pi_u^{(M)}(v), u \rangle_M = 0.$$

In this case, the residual is exactly orthogonal (under $M$) to the projection direction.

## F. Practical Notes on Degenerate Metrics and Stabilization

Our dictionary-induced matrix $M = W_\theta^\top W_\theta$ is always symmetric positive semi-definite. In practice, $W_\theta$ may be rank-deficient, so $M$ can be singular and the induced geometry may be degenerate along $\text{Null}(W_\theta)$. This does not prevent our residual inference from being well-defined, since we use the stabilized coefficient

$$\alpha = \frac{\langle v, \hat{v} \rangle_M}{\|\hat{v}\|_M^2 + \varepsilon}, \qquad \varepsilon > 0,$$

which guarantees a strictly positive denominator. Empirically, we choose a small $\varepsilon$ to ensure numerical stability while keeping the $M$-orthogonality deviation negligible.

*Table 5.* Notations used in `ExtraCare`.

| Symbol | Description |
|---|---|
| $\mathcal{D} = \{(x^{(i)}, y^{(i)})\}_{i=1}^n$ | EHR dataset with $n$ patients. |
| $x^{(i)} = \{V_1^{(i)}, \ldots, V_T^{(i)}\}, V_t^{(i)}$ | Multi-visit record of patient $i$; the $t$-th visit (admission/encounter). |
| $\mathcal{C}, x \in \mathbb{R}^{T \times |\mathcal{C}|}$ | Medical code vocabulary; visit-sequence input encoded over $\mathcal{C}$. |
| $y \in \mathbb{R}^o$ | Ground-truth label vector; $o$ is the output dimension. |
| $\mathcal{D}_s = \{(x_i, y_i)\}_{i=1}^{n_s}, \mathcal{D}_t = \{x_i'\}_{i=1}^{n_t}$ | Source (labeled) and target (unlabeled) domains. |
| $P_s, P_t, w(x)$ | Source/target data distributions; density ratio for covariate shift. |
| $f_\phi(\cdot), v = f_\phi(x), v' = f_\phi(x')$ | Feature encoder and patient representations for source/target samples. |
| $p_\varsigma(\cdot), \hat{y} = p_\varsigma(v)$ | Label prediction head and predicted outcome. |
| $\ell_{\text{CE}}(\cdot), \text{MMD}(\cdot, \cdot)$ | Cross-entropy loss; Maximum Mean Discrepancy for feature alignment. |
| $L_{\text{label}}, \lambda_1$ | Label prediction objective with MMD regularization; $\lambda_1$ is its weight. |
| $h_\theta(\cdot), W_\theta, W_\theta^\top$ | Sparse autoencoder (SAE) encoder; tied dictionary weights for decoding. |
| $s = h_\theta(v) \in \mathbb{R}^{d_s}, \hat{v} = W_\theta^\top s$ | Sparse latent code and reconstructed representation. |
| $L_{\text{rec}}, \gamma$ | Reconstruction loss and $\ell_1$ sparsity coefficient. |
| $M = W_\theta^\top W_\theta$ | Dictionary-induced latent metric (symmetric positive semidefinite). |
| $\langle a, b \rangle_M = a^\top M b, \|a\|_M^2 = a^\top M a$ | Metric inner product and induced (semi-)norm. |
| $\epsilon$ | Stabilization constant for projection (numerical stability). |
| $\alpha = \dfrac{\langle v, \hat{v} \rangle_M}{\|\hat{v}\|_M^2 + \epsilon}$ | Coefficient of the $M$-weighted (regularized) projection onto $\hat{v}$. |
| $z = v - \alpha \hat{v}$ | Domain-specific residual (covariate feature) inferred from $v$. |
| $\delta \in \{0, 1\}, d_\omega(\cdot)$ | Domain indicator (0: source, 1: target); domain classifier. |
| $L_{\text{dcl}}$ | Domain classification loss trained on $z$ (and $z'$). |
| $\mathbb{E}[\cdot]$ | Expectation operator. |
| $\text{sg}(\cdot)$ | Stop-gradient operator. |
| $\langle \cdot, \cdot \rangle, \|\cdot\|$ | Standard Euclidean inner product and norm. |

## G. Notation Table

For clarity, we provide a comprehensive notation table (as shown in Table 5) summarizing all symbols used throughout the paper. We use plain letters (e.g., $n, T, \alpha, \gamma, \varepsilon$) to denote scalars and hyperparameters. Lowercase letters (e.g., $v, s, z$) represent vectors, including patient representations, sparse latent codes, and domain-specific residual features. Uppercase letters (e.g., $M, W_\theta$) denote matrices, such as the dictionary-induced metric and learnable linear transformations. Calligraphic letters (e.g., $\mathcal{D}, \mathcal{C}$) are used to represent datasets, domains, or concept vocabularies. Functions with subscripts (e.g., $f_\phi, h_\theta, g_\varsigma, d_\omega$)

denote neural network modules with their corresponding parameters. We use $\langle \cdot, \cdot \rangle$ to denote the standard Euclidean inner product, and $\langle \cdot, \cdot \rangle_{\mathbf{M}}$ for the dictionary-induced inner product defined by the sparse autoencoder. Unless otherwise specified, $\| \cdot \|$ denotes the Euclidean norm, while $\| \cdot \|_{\mathbf{M}}$ denotes the norm induced by the metric $\mathbf{M}$.

This notation table is intended to serve as a reference for the main text and appendices, ensuring clarity and consistency across the formulation of feature alignment, sparse reconstruction, and orthogonal residual inference.

## H. Pseudo Code for `ExtraCare`

Since the training and inference phases have been explained in the main paper, we present our pseudocode by these two consecutive phases, as shown in Algorithm 1.

---

**Algorithm 1** Training and Inference for `ExtraCare`

---

*// Training*
**Require:** Source training data $(x, y) \sim \mathcal{D}_s$, unlabeled target data $x' \sim \mathcal{D}_t$
Initialize feature extractor $f_\phi(\cdot)$, label predictor $p_\zeta(\cdot)$, SAE $(h_\theta, W_\theta)$, and domain classifier $d_\omega(\cdot)$
Train $f_\phi(\cdot)$ and $p_\zeta(\cdot)$ on source supervision
**for** each training iteration **do**
    **for** each minibatch $(x, y, x')$ **do**
        Obtain patient representations on source/target by Eq. (2)
        Optimize label prediction with distribution alignment by Eq. (3)
    **end for**
**end for**
**for** each training iteration **do**
    **for** each source minibatch $(x, y)$ **do**
        Encode $v$ by Eq. (2) and reconstruct it with SAE (Sec. 3.3)
        Induce latent metric $M$ by Eq. (5) and optimize reconstruction objective by Eq. (6)
        Update $(\phi, \zeta, \theta)$ with the combined objective described in Sec. 3.5
    **end for**
**end for**
**for** each training iteration **do**
    **for** each minibatch $(x, x')$ **do**
        Compute residual covariates $z, z'$ by Eq. (7)
        Train domain classifier on residuals by Eq. (8)
        Update $(\phi, \zeta, \theta, \omega)$ with the combined objective described in Sec. 3.5
    **end for**
**end for**
*// Inference*
**Require:** Target testing data $x' \sim \mathcal{D}_t$
**for** each target patient $x'$ **do**
    **// Stage I: Task Prediction on Aligned Feature**
    Obtain representation $v'$ by Eq. (2)
    Predict outcome $\hat{y}'$ by the label predictor $p_\zeta(\cdot)$ (Sec. 3.1)
    **// Stage II: Sparse Concept Decomposition**
    Obtain sparse concept vector $s$ by the SAE encoder $h_\theta(\cdot)$ (Sec. 3.3)
    Reconstruct aligned feature $\hat{v}$ and metric $M$ by Eq. (4) and (5)
    **// Stage III: Orthogonal Covariates & Interpretation**
    Infer covariate residual $z$ by Eq. (7)
    Perform sparse-dimension ablation on top activated concepts and compute $\Delta\text{prob}(c)$ following Appendix I.1
**end for**

---

# I. Additional Experimental Setup & Results

## I.1. Clinical Predictive Tasks & Evaluation Metrics

The model can be adapted for a variety of healthcare prediction tasks, and we use two representative diagnostic tasks to predict health events:

- **Diagnosis Prediction.** This task predicts all medical codes, i.e. diagnoses of the visit $T + 1$ given previous $T$ visits, which is a multi-label classification.

- **Heart Failure (HF) Prediction.** This task predicts whether a patient will be diagnosed with heart failure in the visit $T + 1$ given previous $T$ visits, which is a binary classification.

For both tasks, a fully connected layer with a sigmoid activation function and a binary cross-entropy loss are adopted as the predictive head and the objective function, respectively. The evaluation metrics for diagnosis prediction are weighted F1 score w-$F_1$ and top-k recall (R@k). w-$F_1$ is a weighted sum of the F1 score for each class, which measures an overall prediction performance on all classes. R@k is the ratio of true-positive numbers in top-k predictions to the total number of positive samples, which measures the prediction performance on a subset of classes. The metrics to evaluate the HF prediction are the area under the ROC curve (AUROC) and $F_1$ score due to the imbalanced label distribution.

## I.2. Implement Details of `ExtraCare`

Considering the common use of the Encoder-Decoder structure for clinical prediction, we adopt the Transformer (Vaswani et al., 2017; Tong et al., 2025a) as the backbone extractor $f_\phi$ in `ExtraCare` and all baselines. Specifically, we follow the implementation adapted from PyHealth (Yang et al., 2023b), consisting of three layers with a hidden size of 64, 4 attention heads, and a dropout rate of 0.2. The position encoding is applied across patient visits to capture temporal order. Only diagnosis codes are used as input features: codes are embedded via an embedding look-up table, summed within each visit, and then passed to the Transformer. For pooling, we use the representation at the first visit index returned by each Transformer block; per-feature outputs are concatenated and fed to a linear layer to produce a 128-dimension embedding. Given backbone features $v \in \mathbb{R}^d$, we apply a sparse autoencoder (SAE) with latent dimension 256 ($k = 2d$). The SAE uses a linear encoder with ReLU activation, tied decoder weights, reconstruction loss, and L1-norm regularization. Domain-specific features $z$ are obtained by an orthogonal projection using the SAE decoder weights. The domain classifier is an MLP with hidden sizes 256 and 128, ReLU activations, and dropout 0.3, followed by a 2-way softmax for source/target prediction.

Default hyperparameters are: embedding dim 128, latent dim 256, heads 4, dropout 0.1, Transformer layers 3, batch size 128, and 30 epochs. Loss weights are $\lambda_{\text{label}} = 1$, $\lambda_{\text{rec}} = 5e^{-3}$, and $\lambda_{\text{dcl}} = 0.3$. The MMD alignment uses a multi-kernel Gaussian kernel with kernel_mul $= 2.0$ and kernel_num $= 5$. The binary cross-entropy with logit loss is used for the supervised label supervision. Training is conducted with a two-stage schedule: first epochs 1-10 update parameters by optimizing the weighted sum of label prediction and reconstruction losses, and epochs 11-30 involves domain supervision for optimization. We sample 500 unlabeled target patients from the validation set; source batches are drawn from the training split, and target batches are drawn from the sampled test subset. We optimize all parameters jointly using Adam with an initial learning rate of $1 \times 10^{-3}$. A multi-step LR scheduler decays the learning rate to $\{10^{-4}, 10^{-5}, 10^{-6}\}$ at epochs $\{15, 20, 25\}$ for the multi-label task, or at $\{10, 20, 25\}$ for HF prediction. We select the best checkpoint based on the highest test-set F1 score observed during training (checkpoint saved each epoch). All experiments are conducted using Python 3.10 and PyTorch 2.3.1 with CUDA 12.4 on a server equipped with AMD EPYC 9254 24-Core CPUs and NVIDIA L40S GPUs. [1]

## I.3. Dataset

We conduct experiments on two real-world EHR datasets, which differ a lot in scale and temporal span settings.

- The eICU dataset (Pollard et al., 2018) includes over 200K visits from 139K patients admitted to ICUs across 208 hospitals in the United States between 2014 and 2015. Hospitals are grouped into four geographic regions: Midwest, Northeast, West, and South. Since hospital visit timestamps are unavailable, each hospital visit is treated as an independent patient instance, and each ICU admission is considered a separate admission record. We extract 20,227 visits from 9,408 patients after preprocessing.

---

[1] https://github.com/humphreyhuu/ExtraCare

*Table 6.* Dataset statistics (Overall & Temporal).

| Item | eICU | OCHIN |
|---|---|---|
| **Overall** | | |
| # patients | 9,408 | 199,703 |
| Max. # visits (encounters) | 7 | 51 |
| Avg. # visits (encounters) | 2.15 | 13.42 |
| Max. # diseases / visit | 54 | 31 |
| Avg. # diseases / visit | 4.40 | 2.89 |
| Most frequent codes | 518.81 (30.68%); 584.9 (18.22%); 401.9 (18.18%); 486 (17.92%); 427.31 (16.56%) | I10 (48.62%); F41.9 (23.95%); F41.1 (21.73%); F17.200 (10.10%); F33.1 (9.21%) |
| # hospitals (facilities) | 187 | 2,936 |
| **Year: 2014** | | **Year: 2012–2021** |
| # patients | 4,687 | 78,786 |
| Max. # visits (encounters) | 7 | 51 |
| Avg. # visits (encounters) | 2.16 | 9.62 |
| Max. # diseases / visit | 54 | 25 |
| Avg. # diseases / visit | 4.56 | 2.61 |
| Most frequent codes | 518.81 (31.51%); 401.9 (19.18%); 486 (17.94%); 584.9 (17.64%); 427.31 (16.77%) | I10 (39.68%); F41.9 (18.93%); F41.1 (15.45%); A311 (14.25%); F419 (12.15%) |
| # hospitals (facilities) | 168 | 1,948 |
| **Year: 2015** | | **Year: 2022–2023** |
| # patients | 4,721 | 120,917 |
| Max. # visits (encounters) | 6 | 51 |
| Avg. # visits (encounters) | 2.14 | 15.89 |
| Max. # diseases / visit | 47 | 31 |
| Avg. # diseases / visit | 4.24 | 3.00 |
| Most frequent codes | 518.81 (29.85%); 584.9 (18.79%); 486 (17.90%); 401.9 (17.18%); 427.31 (16.35%) | I10 (54.44%); F41.9 (27.21%); F41.1 (25.83%); F17.200 (11.19%); F33.1 (11.04%) |
| # hospitals (facilities) | 166 | 2,666 |

- The OCHIN dataset (DeVoe & Sears, 2013) is a large-scale longitudinal EHR repository based on a shared Epic system, covering more than 8.6M patients across approximately 2,400 clinical institutions in over 40 U.S. states from 2012 to 2023. Patient visits are defined by medical encounters, and diagnoses (ICD-10-CM codes) are used to construct admission records. Institutions are categorized by either geographic state or facility type (e.g., primary care, family practice). After preprocessing, 577,142 visits from 199,703 patients are retained.

For both datasets, we filter out visits from patients younger than 18 or older than 89 years, visits lasting longer than 10 days, and visits with fewer than 3 or more than 256 timestamps. In addition, eICU visits shorter than 12 hours are excluded, as predictions are made 12 hours after ICU admission. For OCHIN, due to the long temporal span and large number of encounters per patient, only the most recent 50 encounters are retained as medical history, with predictions generated at discharge. For diagnosis prediction tasks, we focus on medical codes that appear in multiple visits. For HF prediction, positive samples account for 38.5% and 21.7% of visits in eICU and OCHIN, respectively. Detailed temporal and spatial statistics of both datasets are reported in Table 6 and Table 7.

## I.4. Data Split

The eICU dataset comprises data collected two-year admission records from over 200 hospitals across the United States, while the OCHIN dataset spans a period of ten years from about 3,000 facilities. Based on statistic results in Table 6, we can also observe that there is no significant difference between admissions in 2014 and 2015. Therefore, we utilize both eICU and OCHIN dataset to evaluate the model's performance across spatial gaps, and the OCHIN dataset only to assess its

performance across temporal gaps.

For the eICU dataset, we divide it into four spatial groups based on regions: Midwest, Northeast, West, and South. Each group is then split into 70% for training, 10% for validation, and 20% for testing. We evaluate the gap between groups by comparing the performance of the backbone model trained on data from within the same group and data from outside the group. The target testing data is selected as the group (Midwest) that exhibits the largest performance gap, while the remaining groups (Northeast, West, and South) are used as the source training data.

Regarding the OCHIN dataset, we first observe that most patient records are collected in either 2022 and 2023 in Table 6. We divide all records into four temporal groups: 2012-2015, 2016-2018, 2019-2021, 2022-2023. Each group is also further split into training, validation, and testing sets with a ratio of 70%, 10%, and 20% respectively. We also find that there is also no significant difference across first three temporal groups. Based on these two observations, we consider patients admitted after 2022 as the target testing data, while all preceding patients are included in the source training data. Similarly, we also divide clinical institutes by two regions with observable distributional gaps: hospitals located in either California (CA) or Oregon (OR), and hospitals located in either Massachusetts (MA) or North Carolina (NC). Such partition is also aligned with human understanding, which belong to the west coast and the east coast, respectively.

### I.5. Baselines

We first compare `ExtraCare` to two naive baselines.

- **Base:** is trained solely on the labeled source-domain data without any domain adaptation strategy. It directly applies the learned model to the target domain and serves as a lower-bound baseline that highlights the severity of domain shift.

- **Oracle:** denotes an upper-bound baseline trained and evaluated directly on labeled target-domain data, assuming full access to supervision in the target domain. It reflects the best achievable performance without domain shift and is mainly used for reference rather than practical deployment.

Both baselines share the same Transformer backbone and prediction heads as other methods, differing only in the data used during training. We then compare `ExtraCare` to both classic and recent domain generalization methods. For a fair comparison, all the methods below are trained on the source training set, selected on the source validation set, and tested on the target testing set.

- **Domain-Adversarial Neural Network (DANN):** learns domain-invariant representations through adversarial training between a feature extractor and a domain discriminator. A gradient reversal layer is introduced so that the feature extractor is optimized to confuse the domain discriminator while remaining discriminative for the source-label prediction task. By minimizing label prediction loss on the source domain and maximizing domain classification loss, DANN implicitly aligns feature distributions across domains.

- **Rethinking Maximum Mean Discrepancy (RMMD):** revisits MMD-based domain adaptation from a theoretical perspective and reveals that regular MMD minimization can degrade feature discriminability. It introduces discriminative regularization strategies that control intra-class compactness and inter-class separability during distribution alignment. The method operates entirely in the latent feature space and does not depend on a specific network architecture.

- **Robust Spherical Domain Adaptation (RSDA):** performs domain adaptation in a spherical feature space by normalizing embeddings onto a hypersphere. Both the classifier and domain discriminator are reformulated as spherical neural networks, enabling adversarial alignment that is less sensitive to feature norm variations. It also introduces a pseudo-labeling mechanism based on a Gaussian–uniform mixture model to mitigate noise in target-domain predictions.

- **Bayesian Uncertainty Alignment (BUA):** incorporates uncertainty estimation into unsupervised domain adaptation. By modeling epistemic uncertainty through Bayesian neural networks, it aligns source and target domains not only in terms of feature distributions but also in uncertainty space. The key insight is that target-domain samples exhibiting higher uncertainty indicate a larger domain gap, and reducing this uncertainty gap facilitates more reliable adaptation.

- **Domain-Agnostic Learning (DAL):** focuses on disentangling class-relevant, domain-specific, and class-irrelevant factors in the latent space. It aims to isolate domain-invariant, label-informative features while suppressing nuisance variations. Its disentanglement design makes it particularly suitable for complex real-world data distributions.

*Table 7.* Dataset statistics (Spatial).

| Item | eICU | OCHIN |
|---|---|---|
| | **Region: South** | **Region: CA** |
| # patients | 3,153 | 59,272 |
| Max. # visits (encounters) | 6 | 51 |
| Avg. # visits (encounters) | 2.13 | 12.88 |
| Max. # diseases / visit | 32 | 31 |
| Avg. # diseases / visit | 4.50 | 2.43 |
| Most frequent codes | 518.81 (31.81%); 401.9 (23.76%); 584.9 (19.70%); 427.31 (17.60%); 486 (16.62%) | I10 (53.56%); F41.9 (23.72%); F41.1 (16.17%); F17.200 (8.21%); A311 (6.81%) |
| # hospitals (facilities) | 54 | 874 |
| | **Region: Northeast** | **Region: OR** |
| # patients | 913 | 38,404 |
| Max. # visits (encounters) | 6 | 51 |
| Avg. # visits (encounters) | 2.22 | 14.90 |
| Max. # diseases / visit | 54 | 24 |
| Avg. # diseases / visit | 8.68 | 2.96 |
| Most frequent codes | 790.6 (38.23%); 263.9 (36.36%); 518.81 (35.60%); 276.52 (27.38%); 401.9 (27.05%) | I10 (38.10%); F41.9 (24.21%); F41.1 (20.16%); A311 (16.14%); F419 (14.01%) |
| # hospitals (facilities) | 12 | 422 |
| | **Region: Midwest** | **Region: MA** |
| # patients | 2,758 | 17,280 |
| Max. # visits (encounters) | 6 | 51 |
| Avg. # visits (encounters) | 2.13 | 17.35 |
| Max. # diseases / visit | 22 | 25 |
| Avg. # diseases / visit | 3.58 | 3.08 |
| Most frequent codes | 518.81 (28.46%); 486 (19.80%); 584.9 (17.69%); 038.9 (16.57%); 427.31 (15.99%) | I10 (47.69%); F41.9 (25.13%); F41.1 (19.80%); F43.10 (11.88%); F33.1 (10.94%) |
| # hospitals (facilities) | 62 | 208 |
| | **Region: West** | **Region: NC** |
| # patients | 1,945 | 10,661 |
| Max. # visits (encounters) | 6 | 51 |
| Avg. # visits (encounters) | 2.16 | 12.96 |
| Max. # diseases / visit | 27 | 24 |
| Avg. # diseases / visit | 3.64 | 3.02 |
| Most frequent codes | 518.81 (31.16%); 038.9 (18.87%); 584.9 (16.71%); 401.9 (15.01%); 427.31 (14.76%) | I10 (67.36%); F41.1 (24.91%); F41.9 (21.47%); F17.200 (13.24%); F33.1 (9.30%) |
| # hospitals (facilities) | 40 | 122 |

- **Recursively Conditional Gaussian (RCG):** is an ordinal-aware unsupervised domain adaptation method designed for tasks with ordered labels, such as medical diagnosis. It introduces a Gaussian prior to enforce ordinal constraints in the class-related latent space. By disentangling ordinal content factors from domain-specific variations, it aligns domains while preserving label order structure.

- **Cycle Self-Training (CST):** addresses the unreliability of pseudo-labels under domain shift by introducing a bidirectional training cycle. In the forward step, pseudo-labels are generated for target samples using a source-trained classifier. In the reverse step, a target classifier trained on pseudo-labeled target data is required to perform well on source-domain samples by updating shared representations. This cyclic constraint encourages pseudo-labels that generalize across domains rather than overfitting to biased target predictions.

- **Safe Self-Refinement for Transformer-based Domain Adaptation (SSRT):** is designed for Transformer backbones. It refines the model using consistency regularization between predictions of original and perturbed latent token

representations from target data. To prevent model collapse under domain shifts, a safe training mechanism dynamically monitors prediction diversity and restores earlier model states when necessary. The method combines adversarial adaptation with safe self-training, making it well-suited for high-capacity models such as Transformers.

Note that RSDA and RCG (feature alignment baselines) also incorporate self-training phases on the target domain, and we remove these components to ensure a fair comparison, and only retain their core representation-level domain adaptation modules applied to Transformer-derived patient embeddings.

**Implementation on HF Prediction.** We unify all baselines by using the same Transformer backbone to encode each sequential record into a patient embedding, and only modify the adaptation modules to operate on this embedding. ① DANN performs adversarial alignment by adding a domain discriminator on top of the embedding with gradient reversal. ② RMMD aligns source/target feature distributions via (class-wise) MMD-style discrepancy regularization that is computed from predicted/true HF labels. ③ RSDA first projects embeddings onto a hypersphere (via normalization) and then conducts adversarial alignment in spherical space, optionally using robust pseudo-label weighting based on embedding-to-prototype distances. ④ BUA is adapted by extracting its core idea—uncertainty alignment in latent space—estimating epistemic uncertainty of patient embeddings through stochastic sampling (e.g., MC dropout) and minimizing the source–target uncertainty gap alongside feature alignment. ⑤ DAL adds a disentanglement layer over the patient embedding to separate task-relevant and nuisance factors, and performs domain alignment primarily on the task-relevant subspace. ⑥ RCG can still be applied in HF because a binary label can be viewed as an ordinal case with $K = 2$, making its ordinal latent prior and self-training mechanism well-defined. ⑦ CST and ⑧ SSRT both leverage target pseudo-labels to refine the classifier: CST uses its forward/reverse cycling between source-trained pseudo-labeling and reverse consistency on the shared representation, while SSRT enforces prediction consistency under perturbations of Transformer token/embedding representations with a safety mechanism to avoid collapse; both are straightforward in binary classification.

**Implementation on Diagnosis Prediction.** We keep the same Transformer backbone but replace the prediction head with a multi-label sigmoid layer and optimize with per-label binary cross-entropy. ① DANN remains directly applicable by aligning embeddings adversarially regardless of output dimensionality. ② RMMD extends by computing class-conditional alignment per label (each label as an independent binary task) to estimate conditional statistics. ③ RSDA similarly extends by defining spherical prototypes and robust pseudo-label weights per label. ④ BUA also transfers by aligning embedding uncertainty for each label prediction using stochastic sampling, and DAL remains compatible by disentangling task-relevant factors from nuisance factors and aligning the task-relevant subspace. ⑤ RCG is not applicable to multi-label classification because it is designed for single-label ordinal classification and the core ordinal constraint is ill-defined for this task. Likewise, ⑥ CST in its original form does not support multi-label diagnosis because it generates target pseudo-labels via an argmax over a softmax multi-class classifier, which conflicts with the required multi-hot label structure. ⑦ SSRT can be adapted by applying consistency regularization and safe self-refinement to the multi-label sigmoid outputs (or intermediate embeddings), as it does not intrinsically assume a single-label softmax formulation.

### I.6. Additional Analysis on Spatial Adaptation in OCHIN

In OCHIN, we define facilities in either California or Oregon (1,296 involved) as the source data, and facilities in either Massachusetts or North Carolina (330 involved) as the target data. Table 8 reports additional domain adaptation results on the OCHIN dataset under spatial shifts defined by cross-state facility groups. Consistent with the main results, a clear performance gap is observed between Oracle and Base across both diagnosis prediction and heart failure prediction, confirming the presence of substantial spatial domain shifts in this setting. Feature adaptation methods consistently improve over Base, with RMMD, RSDA, BUA, and RCG demonstrating stronger performance among alignment-based baselines, while DANN and DAL yield more limited gains. Self-training methods also outperform Base but exhibit higher variance and less consistent improvements across tasks. Overall, `ExtraCare` achieves the best performance across all metrics, remaining close to the Oracle upper bound and consistently outperforming competing methods. These results further support the robustness of our approach under spatial shifts in large-scale, heterogeneous outpatient EHR data.

## J. Detailed Discussion of Interpretation

This appendix provides a detailed discussion of our case study in Figure 2, including (i) the exact ablation and mapping procedure that produces $\Delta$prob, and (ii) clinical interpretation of the selected sparse dimensions for the two representative

*Table 8.* **Results of domain adaptation across spatial gaps on the OCHIN datasets.** We report the average performance (%) and the standard error (in bracket).

| Model | Diagnosis | | Heart Failure | |
|---|---|---|---|---|
| | w-$F_1$ | R@5 | AUROC | F1 |
| Oracle | 76.90 (0.12) | 83.15 (0.07) | 96.31 (0.18) | 87.10 (0.09) |
| Base | 62.89 (0.08) | 76.15 (0.21) | 90.13 (0.11) | 74.19 (0.14) |
| DANN | 67.36 (0.16) | 75.35 (0.10) | 93.45 (0.23) | 76.98 (0.06) |
| DAL | 64.91 (0.05) | 77.38 (0.19) | 92.74 (0.20) | 76.82 (0.12) |
| RMMD | 70.31 (0.09) | 79.16 (0.17) | 91.38 (0.24) | 81.86 (0.08) |
| CST | / | / | 91.88 (0.24) | 77.26 (0.16) |
| SSRT | 69.48 (0.41) | 76.80 (0.32) | 92.66 (0.25) | 79.84 (0.20) |
| RSDA | 69.32 (0.22) | 81.16 (0.13) | 95.29 (0.10) | 82.01 (0.15) |
| BUA | 72.36 (0.07) | 80.28 (0.26) | 93.53 (0.09) | 79.01 (0.20) |
| RCG | / | / | 93.30 (0.16) | 81.01 (0.11) |
| ExtraCare | 74.97 (0.06) | 83.44 (0.12) | 94.52 (0.19) | 84.77 (0.07) |

patients. Throughout this section, we denote the sparse concept vector as $\mathbf{s} \in \mathbb{R}^K$.

### J.1. Sparse-Dimension Ablation and Condition Mapping

**Selecting sparse dimensions.** For each patient $i$, our model produces an encoded patient representation and its sparse concept activations $s^{(i)}$. We select the top activated non-zero sparse dimensions by magnitude:

$$\mathcal{K}^{(i)} = \text{Top-}3 \left( \left\{ |s_k^{(i)}| \right\}_{k=1}^K \right), \tag{14}$$

where $\mathcal{K}^{(i)}$ contains the three indices with the highest activation values (among non-zero entries).

**Ablation protocol.** For each selected dimension $k \in \mathcal{K}^{(i)}$, we construct an ablated sparse vector by setting the $k$-th activation to zero:

$$\tilde{\mathbf{s}}^{(i,k)} = \mathbf{s}^{(i)} - s_k^{(i)} \mathbf{e}_k, \tag{15}$$

where $\mathbf{e}_k$ is the one-hot basis vector. We then feed $\tilde{\mathbf{s}}^{(i,k)}$ into the diagnosis prediction head to obtain the ablated prediction probabilities over all ICD10-CM codes:

$$\tilde{p}^{(i,k)} = f_{\text{clf}} \left( \tilde{s}^{(i,k)} \right), \quad p^{(i)} = f_{\text{clf}} \left( s^{(i)} \right). \tag{16}$$

**Probability change $\Delta$prob.** For each diagnosis label (ICD10-CM code) $c$, we define the ablation-induced probability change as:

$$\Delta\text{prob}^{(i,k)}(c) = \left| p^{(i)}(c) - \tilde{p}^{(i,k)}(c) \right|. \tag{17}$$

In our case study visualization, we use $\Delta\text{prob} = 0.05$ as a practical threshold to distinguish *high label impact* codes (above threshold) from *low label impact* codes (below threshold).

**Domain sensitivity annotation.** In addition to label impact, we annotate each selected sparse dimension as either **domain-sensitive** or **domain-insensitive**. Since the numeric domain probability changes can be unstable in practice, we follow a rank-based criterion in the case study:

- **Domain-sensitive:** top-5 ranked codes by domain impact for that dimension;

- **Domain-insensitive:** bottom-5 ranked codes by domain impact for that dimension.

This rank-based design supports a robust qualitative separation of domain-related variation without relying on potentially noisy magnitude values.

## J.2. Clinical Interpretation: Patient-Level Sparse Dimensions

We provide a qualitative discussion for the two representative patients used in the case study (Figure 2). Our goal is not to claim clinical causality from a single dimension, but to show that the sparse dimensions produce coherent and clinically plausible diagnosis attributions, while highlighting which concepts are more likely to reflect domain-specific variation.

### J.2.1. PATIENT 1: TWO DOMAIN-SENSITIVE CONCEPTS AND ONE DOMAIN-INSENSITIVE CONCEPT

The first patient exhibits three interpretable sparse dimensions:

**Dim 227 (domain-sensitive): anxiety/depression-related concept.** This dimension primarily maps to mental health conditions such as anxiety and depressive disorders (e.g., `F41.1`, `F33.1`). Clinically, these conditions often co-occur and exhibit longitudinal persistence, making them strong predictors in chronic-care EHR trajectories. However, the same concept can also be domain-sensitive in practice, as mental health diagnoses are known to be influenced by heterogeneous screening intensity, access-to-care patterns, and coding preferences across clinics. Therefore, identifying Dim 227 as domain-sensitive is clinically plausible: it remains predictive, yet the way it manifests in structured EHR may shift across cohorts.

**Dim 149 (domain-sensitive): depressive subtype and comorbidity structure.** Dim 149 captures a closely related but more fine-grained subgroup of depression/anxiety codes (e.g., `F33.1`, `F33.0`, `F41.9`), which often form a comorbidity cluster in outpatient records. The ablation results suggest that only a small subset of codes exceed the $\Delta\text{prob} = 0.05$ threshold, which aligns with the interpretation that a sparse concept dimension tends to be dominated by a few clinically central diagnoses rather than an unstructured mixture. This dimension is also domain-sensitive, reflecting that depressive subtyping and documentation practices can differ substantially across institutions and time.

**Dim 5 (domain-insensitive): stable psychiatric baseline evidence.** Dim 5 is identified as domain-insensitive in our case study. Compared with domain-sensitive dimensions, its mapped codes show weaker domain-driven variation and thus can be interpreted as more transferable evidence. In practice, such dimensions provide a form of "stable clinical anchor" for adaptation: they support prediction while being less affected by cohort-specific documentation shifts.

### J.2.2. PATIENT 2: ONE DOMAIN-SENSITIVE CONCEPT AND TWO DOMAIN-INSENSITIVE CONCEPTS

For the second patient, we analyze three selected sparse dimensions:

**Dim 24 (domain-sensitive): mixed cardiometabolic and psychiatric evidence.** Dim 24 contains a mixture of cardiometabolic (`I10`) and psychiatric-related codes (e.g., `F20.4`, `F41.9`). This pattern is clinically plausible because multimorbidity is common in EHR cohorts, and hypertension often coexists with mental health conditions through shared risk factors and treatment pathways. At the same time, the mixture also highlights why domain sensitivity matters: comorbidity profiles and coding emphasis may vary between care settings. Thus, Dim 24 serves as an example where the concept is strongly predictive but must be audited for domain-specific instability.

**Dim 151 (domain-insensitive): transferable predictive evidence under shift.** Dim 151 is categorized as domain-insensitive. Although it can still affect predicted probabilities for certain psychiatric codes, its domain impact is weaker (rank-based), making it more transferable across cohorts. This aligns with our motivation that robust clinical prediction requires retaining stable and generalizable evidence rather than overfitting to cohort-specific artifacts.

**Dim 145 (domain-insensitive): clinically meaningful and stable sparse attribution.** Dim 145 is also domain-insensitive for Patient 2. Notably, some ICD10-CM codes (e.g., `F41.9`) may appear under multiple dimensions across different sensitivity categories. This does not indicate inconsistency; instead, it suggests that a diagnosis code can be activated under multiple latent concepts, where each concept dimension can exhibit a different degree of domain sensitivity. Such behavior is expected in EHR data, where a single diagnosis (e.g., anxiety disorder) can serve as stable predictive evidence in one context, while reflecting domain-dependent coding patterns in another.

## J.3. Understanding the Four-Quadrant Categorization

To support model auditing in domain adaptation, we summarize how the mapped ICD10-CM codes can be organized into four categories based on *label impact* (using $\Delta\text{prob}$ thresholding) and *domain sensitivity* (rank-based):

1. **High label impact & Low domain sensitivity:** transferable predictive evidence that is likely robust across domains.

2. **High label impact & High domain sensitivity:** predictive but shift-sensitive concepts that require careful auditing.

3. **Low label impact & High domain sensitivity:** domain discrepancy indicators that reflect cohort-specific variation without dominating prediction.

4. **Low label impact & Low domain sensitivity:** factors with limited relevance to both prediction and domain shift.

This categorization complements standard interpretability by explicitly revealing which clinically meaningful conditions are most likely to remain stable under domain shift and which may require additional caution.

### J.4. Practical Implications for Clinical Deployment

Our case study highlights two deployment-relevant insights. First, sparse-dimension ablation provides a concept-grounded way to localize predictive evidence to a small set of clinically coherent codes, enabling clinician-facing review. Second, domain sensitivity annotation helps distinguish transferable evidence from shift-sensitive variation, which is critical for safe deployment under temporal or cross-institutional shifts. These properties support transparent adaptation: clinicians can inspect *what evidence transfers* and *what evidence may shift* without treating domain adaptation as a black-box procedure.

### J.5. External Grounding Protocol

This section provides the full protocol for the RQ3 validation summarized in the main text. The study follows the facility adaptation setting: the source cohort is Primary Care/Family Practice, and each target cohort corresponds to a specialized facility. The goal is to test whether the label and domain axes in the explanation are externally grounded, by comparing their induced audit groups against independently defined patient- and facility-level anchors.

**Compared explanation methods.**  We compare four methods: SHAP, Integrated Gradients (IG), regular SAE, and `ExtraCare`. SHAP and IG provide dense attribution scores for observed ICD-10 codes. We compute separate attributions with respect to the label prediction head and the domain classifier, producing one label-related and one domain-related score for each candidate code. For regular SAE, we use sparse code scores derived from the invariant and covariate reconstruction components. `ExtraCare` uses the same sparse concept basis as the main interpretation pipeline, together with the label-impact and domain-sensitivity axes defined in Appendix J.1. Because dense methods can score many more observed codes than SAE-based explanations, we fix the number of candidate codes per patient before audit groups are formed. Specifically, for each patient and each method, candidate codes are ranked by their absolute explanation scores and truncated to the same budget before high/low label and high/low domain partitions are formed. This prevents a method from obtaining an advantage simply by returning longer code lists.

**Audit-group construction.**  For each patient $i$, methods assign candidate codes to four groups:

$$Q_1(i): \text{high-label/high-domain}, \quad Q_2(i): \text{high-label/low-domain},$$
$$Q_3(i): \text{low-label/high-domain}, \quad Q_4(i): \text{low-label/low-domain}.$$

The intended interpretation is that the label axis captures patient-specific predictive relevance, whereas the domain axis captures target-facility specificity. This yields four expected roles: $Q_1$ should contain codes that are both patient-relevant and facility-specific; $Q_2$ should contain transferable patient-relevant codes; $Q_3$ should contain facility-specific but less label-driving codes; and $Q_4$ should contain codes that are weak along both axes.

**External anchors.**  For patient relevance, we define

$$R_i(c) = \mathbb{I}[c \in Y_i \cup H_i], \tag{18}$$

where $Y_i$ is the future diagnosis set and $H_i$ is the historical diagnosis set for patient $i$. This anchor does not claim that a code is causally responsible for the model decision; instead, it checks whether high-label groups are enriched with codes that are clinically grounded in the patient's observed trajectory. For facility specificity, we define

$$D_t(c) = \frac{\max(p_t(c) - p_{\text{src}}(c), 0)}{p_t(c) + p_{\text{src}}(c) + \epsilon}, \tag{19}$$

where $p_t(c)$ is the prevalence of code $c$ in target facility $t$, $p_{\text{src}}(c)$ is the prevalence in the Primary Care/Family Practice source cohort, and $\epsilon$ is a small stabilizer. Larger $D_t(c)$ indicates that a code is more enriched in the target facility relative to the source cohort.

**Axis and audit-group metrics.** We compute three validation metrics, and we define the Label-Axis Difference (LAD) as

$$\text{LAD} = \mathbb{E}[R_i(c) \mid c \in Q_1(i) \cup Q_2(i)] - \mathbb{E}[R_i(c) \mid c \in Q_3(i) \cup Q_4(i)], \tag{20}$$

which tests whether high-label groups contain more patient-relevant codes than low-label groups. Similarly, we define the Domain-Axis Difference (DAD) as

$$\text{DAD} = \mathbb{E}[D_t(c) \mid c \in Q_1(i) \cup Q_3(i)] - \mathbb{E}[D_t(c) \mid c \in Q_2(i) \cup Q_4(i)], \tag{21}$$

which tests whether high-domain groups contain more facility-enriched codes than low-domain groups. Finally, we define the Quadrant Calibration Error (QCE) as

$$\text{QCE} = \frac{1}{4} \sum_{k=1}^{4} \|\mu_{Q_k} - v_{Q_k}\|_2, \tag{22}$$

where $\mu_{Q_k} = (\bar{R}_{Q_k}, \bar{D}_{Q_k})$ is the empirical centroid of audit group $Q_k$, and the ideal anchors are $v_{Q_1} = (1,1)$, $v_{Q_2} = (1,0)$, $v_{Q_3} = (0,1)$, and $v_{Q_4} = (0,0)$. All expectations and centroids are computed over patient-code pairs in the corresponding target facility, with $t$ denoting the facility of patient $i$. Higher LAD and DAD indicate clearer axis separation, while lower QCE indicates better overall audit-group calibration.

**Judge protocol and rubric.** After the anchor-based comparison, we conduct a pairwise judge study between regular SAE and `ExtraCare`. We focus on this pair because both methods rely on sparse concepts; the comparison therefore isolates whether our joint label-domain factorization improves explanation organization beyond sparsity alone. For each case, explanations are anonymized and shown in randomized order. The LLM judge evaluates 60 patients from each of the five target facilities (300 cases total). Two human evaluators with medical training evaluate 5 patients from each target facility (25 cases total), and we report averaged scores.

The judge rubric contains three criteria. Quadrant Rule Satisfaction (QRS, 0–4) awards one point for each displayed group whose codes match the intended role described above. Clinical Coherence (1–5) assesses whether codes within the explanation form a medically plausible patient-level summary, such as coherent comorbidity or specialty-specific patterns. Partition Clarity (1–5) assesses whether the label and domain axes are easy to distinguish and whether the four audit groups provide non-redundant information. Preferred (%) is the fraction of paired cases where the judge selects one explanation as the better overall clinical-domain adaptation summary. This rubric is designed to evaluate explanation organization and auditability, not to certify causal correctness of individual code attributions.

## K. Additional Ablation Results

This section extends the ablation study in the main text with two targeted analyses. First, we intervene on the residual component $z$ to test whether it functionally affects target-domain predictions under facility shifts. Second, we examine whether the label-impact threshold used for interpretation is stable around the value used in Figure 2.

### K.1. Residual Masking Intervention

We examine whether the residual component $z$ has a functional role beyond being linearly decodable by the domain classifier. For each target patient in the facility adaptation setting, we first compute the original representation $v$ and its residual decomposition $z = v - \alpha\hat{v}$. We then mask the residual by replacing $v$ with $\tilde{v} = \alpha\hat{v} = v - z$, while keeping the same prediction head $p_\zeta(\cdot)$ fixed. This removes the domain-specific residual at inference time without retraining the classifier. We report the original target-domain R@5, the masked R@5, and the average prediction change $\mathbb{E}_{x,c}[|p_\zeta(v)_c - p_\zeta(\tilde{v})_c|]$, computed over target patients and diagnosis codes.

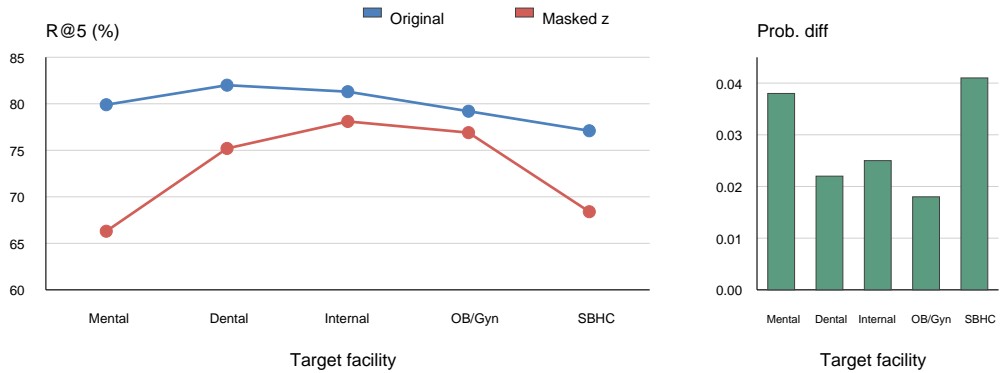

*Figure 5.* **Residual masking under facility shifts.** Removing $z$ consistently reduces target-domain R@5 and induces nontrivial probability changes, with larger effects on Mental Health and SBHC.

Figure 5 shows that masking $z$ consistently reduces retrieval performance across all five target facilities. The largest drops occur in Mental Health (79.9 to 66.3) and SBHC (77.1 to 68.4), which are also the more strongly shifted target cohorts in the facility adaptation analysis. The average probability difference follows the same pattern, suggesting that $z$ carries facility-specific variation actively used by the prediction head. Table 9 further connects the intervention to the audit groups: low-domain codes remain comparatively stable after masking, whereas high-domain codes change more strongly. Thus, the residual is not merely a separable domain signal; intervening on it selectively alters facility-sensitive predictions.

*Table 9.* **Code-level residual masking in Mental Health.**

| ICD-10 | Domain group | $p(v)$ | $p(\tilde{v})$ | $|\Delta|$ |
|--------|--------------|--------|----------------|------------|
| I10 | Low Domain | 0.312 | 0.329 | 0.017 |
| A31.1 | Low Domain | 0.086 | 0.094 | 0.008 |
| F41.9 | High Domain | 0.274 | 0.208 | 0.066 |
| F41.1 | High Domain | 0.251 | 0.159 | 0.092 |

### K.2. Sensitivity to the Label-Impact Threshold

The qualitative interpretation in Figure 2 uses the ablation-induced probability change threshold $\Delta\text{prob} = 0.05$ to separate high-label-impact codes from low-label-impact codes. To assess whether this choice is fragile, we sweep the threshold $\tau$ while keeping the same patients, sparse dimensions, candidate codes, and domain ranking fixed. For each threshold, we recompute the four audit-group centroids and QCE defined in Appendix J.5. This analysis only changes the visualization threshold used for label-impact grouping; it does not change model training or prediction.

*Table 10.* **Sensitivity of audit-group calibration to the label-impact threshold.** The adopted threshold $\tau = 0.05$ is highlighted in gray and yields the lowest QCE.

| $\tau$ | $Q_1$ | $Q_2$ | $Q_3$ | $Q_4$ | QCE $\downarrow$ |
|--------|-------|-------|-------|-------|------------------|
| 0.04 | (0.65, 0.60) | (0.60, 0.44) | (0.43, 0.57) | (0.39, 0.40) | 0.57 |
| 0.05 | **(0.81, 0.78)** | **(0.76,** 0.23) | (0.27, 0.75) | **(0.20, 0.18)** | **0.32** |
| 0.06 | (0.72, 0.67) | (0.65, **0.21**) | **(0.25, 0.78)** | (0.32, 0.29) | 0.40 |
| 0.07 | (0.63, 0.58) | (0.57, 0.47) | (0.45, 0.55) | (0.41, 0.42) | 0.61 |

Table 10 shows that the audit-group structure is stable around $\tau = 0.05$–0.06. The threshold $\tau = 0.05$ yields the lowest QCE (0.32), while $\tau = 0.06$ still places several centroids close to their intended corners but increases the overall calibration error. Larger or smaller thresholds degrade calibration: $\tau = 0.04$ admits more moderately affected codes into the high-label groups, whereas $\tau = 0.07$ over-prunes codes with meaningful but not extreme label impact.

