# OpenReview forum: "Exploring Accurate and Transparent Domain Adaptation in Predictive Healthcare via Concept-Grounded Orthogonal Inference"
_ICML.cc/2026/Conference — ICML 2026 regular_

### Official Review · Reviewer_3wm2 · 2026-03-12

**Soundness:** 3
**Presentation:** 2
**Significance:** 2
**Originality:** 2
**Overall Recommendation:** 4
**Confidence:** 3

**Summary:**

This paper proposes a method of domain adaptation on electronic health records (EHR) using a sparse autoencoder (SAE) to decompose latent representations into a label-dependent feature and domain specific residual. Ablating the sparse vector representation gives insight into the impact of medical codes on probability outcomes and whether they may be domain-sensitive.

**Compliance With Llm Reviewing Policy:**

Affirmed.

**Final Justification:**

The authors have addressed most of my concerns and I raise my score from weak reject to weak accept. As one of the main contributions of the paper is the interpretability aspect, I believe the paper would still benefit from greater clarity regarding the understanding of their model's interpretability.

**Key Questions For Authors:**

1. As mentioned in weaknesses, how does this paper's method differ from Bousmalis et al. (2016) and Sun et al. (2022b)? Is there a reason the authors did not compare against this similar method in their experiments?
2. Have there been any other studies into explainable DA, either in healthcare or in other fields? What is the position of this paper with respect to those?
3. Some interpretability examples have been given for diagnosis prediction in Appendix J.2, but how does interpretability apply to the heart failure prediction task? In particular, what would Figure 2 look like for heart failure prediction?
4. How did the authors validate their model interpretability and its usefulness? Can they provide clinician feedback or a scenario where its use leads to improved patient outcomes? It seems to me that the correlation between label and domain impact will ultimately reflect label shift between domains and that this could be predicted from prior facility knowledge.

The first two questions affect my presentation score and the latter two the significance and originality scores.

**Limitations:**

Limitations of their method are not explicitly discussed. E.g. how sensitive is their method to hyperparameters? How well does the model perform and how is model interpretability affected in distribution shifts affecting class balance?

**Strengths And Weaknesses:**

Strengths:
* Experiments are well-designed and investigate a range of distribution shifts include spatial, temporal and population/facility shift.
* The ablation study showed that each part of the authors' proposed method contributed to their improved performance.
* The paper proposes a novel combination of methods for clinical DA.

Weaknesses:
* Some differences with prior work are unclear. The authors mention domain separation networks (Bousmalis et al., 2016) and follow-up works (Sun et al., 2022b) which similarly orthogonally partition domain-specific and shared subspaces, but the differences between their work and the authors' is not discussed.
* Limitations of the method are not discussed.
* The interpretability of saying that a medical code, which is the label, has different levels of label impact is unclear to me.
* In Figure 2c, the claim that a medical code has different labels of label impact when the label is the medical code itself is not very interpretable to me.
* The significance of the interpretability (Section 4.2) was not very clear despite RQ2 being one of the main contributions of the paper. It may be beneficial to include practical examples and how they may inform clinicians' actions in the main body of the paper.

---

> ### Author Rebuttal · Authors · 2026-03-31
>
> Thanks for your detailed suggestions. Our responses are as follows.
>
> ---
>
> **Q1:** We did not intentionally avoid comparison to these works. Even when methods all split the encoded feature, the splitting mechanism can still be very different. For example, DSN uses separate dense layers to learn different objectives and thus gives a relatively simple approximation of shared/private decomposition. The key design difference in our model is that both the invariant and covariate parts are controlled by sparse activations. Regarding the specific baselines: we did not include Bousmalis et al. (2016) mainly because it is an early method and we prioritized more recent baselines; we did not include Sun et al. (2022b) because its reproduced orthogonality loss is always 0, and its original design is built around U-Net rather than EHR sequence modeling. Moreover, we already select two other same-year studies DANN and SSRT as earlier baselines. We will clarify this more explicitly in the revision.
>
> **Q2:** As discussed in the Introduction, to our knowledge, even in general DA there is still no prior study that jointly explains invariant and covariate information. In computer vision, some studies compare intermediate outputs with the original input image [1,2,3], which is meaningful because features correspond to pixels in the image. However, this does not generalize well to the clinical domain. The semantic information is carried by medical code embeddings, so comparing domain outputs back to the raw input is not meaningful in the same way. Similarly, code embeddings by themselves do not make the model interpretable: methods such as SHAP or LIME can only explain how specific embedding dimension affects the label, which is not human-understandable, and standard SAE can still only explain either the invariant part or the covariate part separately. We will discuss this topic in Related Work for better clarity.
>
> **Q3:** Because of the way SAE is used for interpretation, the learned dictionary is naturally tied to the label space. For diagnosis prediction, the label space is exactly the disease vocabulary, so the resulting interpretation is directly at the disease level. For HF prediction, the label space only contains a single target, so the interpretation would be much more limited if we stayed strictly within that task. One practical extension is to first train the model under the HF task and then learn the SAE and prediction head parameters on diagnosis prediction, so that a similar explanation at the code level can still be obtained. This is also why we view our model as especially suitable for prediction at the code level. We will add this clarification and supplementary explanation in the revision.
>
> **Q4:** Thank you for the insight. To our knowledge, there is currently no prior facility knowledge in the open datasets that would support this type of validation in a clean way. Although eICU is multi-center, it only provides region information and does not include sufficiently institute-level metadata for this purpose. We also note that, as discussed in RQ2, the same clinical concept can serve as robust predictive evidence in some contexts while acting as a domain-sensitive signal in others, and these two roles are not contradictory. Hence, it is difficult to define a fully fair benchmark based on prior facility knowledge alone. We agree this would still be a useful supplementary validation. Our current evidence is initial interpretation, and we think this work can be considered as a fundamental step before constructing comprehensive evaluation intervention metrics. We will clarify these potentials and limitations in the revision.
>
> ---
>
> **Limitation:** Thank you for the reminder. Based on the suggestions from all reviewers, we will add a new section in the revised manuscript to discuss the potential limitations. Please also refer to our response to `Reviewer 2TZk, Limitation` for a more detailed discussion. In general, our paper aims to bring the XAI inspiration to clinical DA rather than to present a fully verified mechanistic explanation method. We fully agree with your concern about the usefulness of interpretation. As with most XAI work, we view interpretability primarily as a tool for observation and analysis rather than as a guarantee of intelligence or causality by itself. It offers insight into the model's reasoning process, but more concrete validation benchmarks in clinical DA remain underexplored. We will make this positioning clearer in the revision.
>
> *Reference:*
>
> *[1] Visualizing adapted knowledge in domain transfer. CVPR 2021.*
>
> *[2] Explaining cross-domain recognition with interpretable deep classifier. ACM Transactions 2023.*
>
> *[3] Learning transferable conceptual prototypes for interpretable unsupervised domain adaptation." IEEE TIP 2024.*
>
> ---
>
> Thanks again for taking the time to review our paper and provide insightful feedback. We are open to discussion on other technical or experimental details.

---

> > ### Author Rebuttal · Reviewer_3wm2 · 2026-04-02
> >
> > Thank you for the response and clarification - my concerns are mostly addressed. I am still unclear about the interpretation of a label having differing label impact as described in weaknesses W3 and W4 however. In the case of a low label impact such as the top left F41.9 in Fig 2c, is it more correct to say that sparse dimension 237 has a low impact on predicting label F41.9? Otherwise it is not clear to me how a label (the medical code) can have low label impact. I believe this point could use a clearer explanation.

---

> > > ### Author Response · Authors · 2026-04-03
> > >
> > > We are glad that our previous response addressed most of your concerns, and we now focus on the remaining issue below.
> > >
> > > ---
> > >
> > > First, we acknowledge that our current shorthand in Fig. 2c may not have made the intended interpretation fully explicit. Formally, “label impact” is not meant to describe an intrinsic property of the medical code itself, and it refers to the ablation-induced effect of a selected sparse dimension on the predicted probability of a given label.
> > >
> > > In the mentioned F41.9 case, your rephrasing is indeed a more explicit way to read the figure: it is more precise to say that **sparse dimension 237 has low impact on predicting F41.9 for that patient**. This futher means that, **within the latent clinical concept captured by sparse dimension 237, F41.9 is not a primary piece of retained predictive evidence for this patient**. In general, the quantity is dimension-conditional and patient-specific, which is also why the same ICD code can appear in different quadrants across different sparse dimensions: the attribution depends on which latent concept is being ablated, rather than on a fixed property of the label itself.
> > >
> > > ---
> > >
> > > **Adds-On Validation:** If the reviewer is still interested in the validation of intrepretation and concept distinction, please see our latest response to `Reviewer 6Gcz, Validation`. There we conduct a new validation analysis with standard XAI methods (Integrated Gradient, SHAP, and SAE), where our method achieves the best quadrant separation (LAD/DAD/QCE: 0.47/0.42/0.32) and is also preferred by both LLM (68%) and human judges (64%). These results can support that our explanations are quantitatively stronger than existing XAI baselines.
> > >
> > > ---
> > >
> > > Thank you again for this detailed example and thoughtful discussion! Due to the one-round rebuttal format, we have done our best to address your concerns as concretely as possible. We will make this point more explicit and avoid this possible ambiguity for readers in the revised manuscript, and we hope this response helps resolve your remaining concerns.

---

### Official Review · Reviewer_sEkE · 2026-03-12

**Soundness:** 2
**Presentation:** 3
**Significance:** 3
**Originality:** 3
**Overall Recommendation:** 4
**Confidence:** 3

**Summary:**

This paper proposes ExtraCare, a domain adaptation framework for clinical prediction on EHR data that aims to improve both robustness under distribution shift and model transparency. It aligns source and target representations, decomposes patient features into label-related and domain-related components. An orthogonality-based design links sparse latent dimensions to medical concepts for interpretation.

**Compliance With Llm Reviewing Policy:**

Affirmed.

**Final Justification:**

Given the overall evaluation of this work I would keep Weak accept.

**Key Questions For Authors:**

1. If the target domain violates the covariate-shift assumption, is M-orthogonality sufficient to support a functional separation between label information and domain-specific variation? Can an orthogonal decomposition in the M-induced geometry necessarily isolate genuinely label-related features? Since the model is trained jointly on labeled source data and unlabeled target data, the learned representations may still retain target-domain distribution discrepancy rather than purely task-relevant information. How to reconcile this possible entanglement between label information and domain-related variability?

2. Section 4.3 performs ablation on the three dimensions with the highest activations. If the ranking of sparse dimensions by activation magnitude changes across target domains, does this top-3 interpretation remain reliable? In particular, when the activation distribution in the target domain differs substantially from that observed during training, how do the authors justify the stability and transferability of the interpretation?

3. A four-quadrant categorization based on label impact and domain sensitivity. Given that this framework is fundamentally a post-hoc interpretive summary, what evidence shows that the model has mechanistically separated transferable predictive evidence from domain-specific variation? Have the authors conducted any experiments that filter features according to quadrant membership and then re-evaluate predictive performance or domain sensitivity?

4. ∆prob = 0.05 chosen as the threshold for distinguishing high label impact from low label impact. Is there an analysis with respect to this threshold?

**Limitations:**

They should acknowledge that concept-grounded explanations do not guarantee causal validity and may create overconfidence if clinicians treat them as mechanistic evidence rather than model-based attributions.

**Strengths And Weaknesses:**

The paper proposes ExtraCare, which further applies SAE reconstruction and orthogonal residual decomposition to aligned representations in domain adaptation, enhancing predictive robustness and concept-level interpretability. The problem formulation is important, and the method combination demonstrates some novelty, with experiments covering eICU and OCHIN datasets showing strong performance. However, the current version still has critical gaps in methodological rigor and experimental design credibility, particularly regarding the threshold and "domain sensitivity" definition in the interpretation module, which are somewhat heuristic and insufficient to support the paper's strong transparency claims.

---

> ### Author Rebuttal · Authors · 2026-03-31
>
> Thanks, we appreciate your recognition of our paper and the constructive feedback. Our detailed responses are as follows.
>
> ---
>
> **Q1:** We agree that if covariate shift is violated, i.e., $P_s(y \mid x) \neq P_t(y \mid x)$, then $M$-orthogonality alone does not yield a theoretically strict semantic decomposition. However, this is the shared gap between theory and practice in unsupervised DA. This is also exactly why quantitative evaluation matters. In our setting, the diagnosis distributions across source and target subsets are not the same, yet ExtraCare still consistently outperforms other DA baselines in Table 2. We will clarify more explicitly that our decomposition is structurally motivated and empirically validated, rather than a universal semantic guarantee.
>
> **Q2:** Thank you for the question. We do not expect the ranking of top dimensions to remain stable across domains. Patients have inherently heterogeneous medical histories, and domain shifts may emphasize different latent clinical concepts in the representation. In our view, this does not undermine interpretability. Instead, it reflects how different concepts become dominant under different contexts, which is itself meaningful. We do observe that some concepts (e.g., mental-health conditions) may frequently appear among the top dimensions, but we cannot claim that such concepts are universally preserved across domains. As discussed in Lines 373-377, the same clinical concept can serve as robust predictive evidence in some contexts while acting as a domain-sensitive signal in others, and these two roles are not contradictory.
>
> **Q3:** We agree that the four-quadrant plot itself is a post hoc interpretive summary. For the decomposition effectiveness, we already provide quantitative results through linear probing in Section 4.5 and Table 3, which test the relationship among encoded, domain, and label information, together with the theoretical support in the method section. For the last question, we agree that an additional validation would further strengthen our claims. However, there is currently no prior facility knowledge in the open datasets that would support this type of validation in a clean way. Although eICU is multi-center, it only provides region information and does not include sufficiently institute-level metadata for this purpose. Hence, it is difficult to define a fully fair benchmark based on prior facility knowledge alone. We think this work can be considered as a fundamental step before constructing comprehensive evaluation intervention metrics. We will clarify these potentials and limitations in the revision.
>
> **Q4:** We did not tune the $\Delta$prob threshold for model performance, since this threshold only affects the illustration of the case study and is not part of training or inference. In practice, we found that ablating a single sparse dimension rarely changes a label probability by more than 0.1, so the relevant range is naturally below 0.1. If the threshold is set much lower than 0.05, the visualization retains too many weakly affected codes and becomes hard to illustrate clearly. For this reason, we used 0.05 as a practical threshold in the paper. We agree that this choice should be explained more clearly, and we will add that explanation in the revision.
>
> ---
>
> **Limitation**: Thank you for the reminder. Based on the suggestions from all reviewers, we will add a new section in the revised manuscript to discuss the potential limitations. Please also refer to our response to `Reviewer 2TZk, Limitation` for a more detailed discussion. In general, our paper aims to bring the XAI inspiration to clinical DA rather than to present a fully verified mechanistic explanation method. We fully agree with your concerns about causal validity and stability. As with most XAI work, we view interpretability primarily as a tool for observation and analysis rather than as a guarantee of intelligence or causality by itself. It offers insight into the model's reasoning process, but more concrete validation benchmarks in clinical DA remain underexplored. We will make this positioning clearer in the revision.
>
> ---
>
> Thanks again for your time and valuable questions. We hope our explanations can clear your concerns. We are happy to explain any other component of the model.

---

> > ### Author Rebuttal · Reviewer_sEkE · 2026-04-01
> >
> > 1. The rebuttal does not directly validate the functional role of the residual ( z ). Its utility could be demonstrated via internal interventions (e.g., masking or perturbing ( z )) to test whether it captures domain-specific variation. Without such evidence, the four-quadrant framework remains insufficiently substantiated.
> >
> > 2. The use of ( \Delta \text{prob} = 0.05 ) for “visual clarity” raises concerns about stability. If small threshold changes alter conclusions, the method may lack robustness for clinical use. A sensitivity analysis is needed to ensure consistency and safety.

---

> > > ### Author Response · Authors · 2026-04-03
> > >
> > > We thank the reviewer for these suggestions. We focus on the remaining concerns with two additional experiments:
> > >
> > > ---
> > >
> > > # Masking $z$
> > >
> > > To directly validate the functional role of the residual z, we performed a residual-masking intervention under the facility-shift setting (see Section 4.6). For each target patient, we first compute the original representation $v$ and its residual decomposition $z=v-\alpha\hat v$, and then replace $v$ with $\tilde v=\alpha\hat v=v-z$ while keeping the same prediction head $p_\zeta(\cdot)$. We report (i) the original target-domain R@5, (ii) the masked R@5 after replacing $v$ by $\tilde v$, and (iii) the average prediction change (Prob Diff) as $\mathbb{E}\_{x,c}\left[\left|p\_\zeta(v)\_c\-p\_\zeta(\tilde v)\_c\right|\right]$.
> > >
> > > |Target facility|Original R@5|Masked R@5|Prob Diff|
> > > |---|---:|---:|---:|
> > > |Mental Health|79.9|66.3|0.038|
> > > |Dental|82.0|75.2|0.022|
> > > |Internal Medicine|81.3|78.1|0.025|
> > > |OB/Gyn|79.2|76.9|0.018|
> > > |SBHC|77.1|68.4|0.041|
> > >
> > > Masking $z$ consistently reduces predictive performance across all five facility shifts. The largest drops appear on Mental Health and SBHC, which are also the most shifted targets in Figure 3. The average probability change follows the same pattern, showing that removing $z$ suppresses facility-specific variation. This provides functional evidence beyond the linear-probe result in Table 3.
> > >
> > > To connect this intervention to our four-quadrant interpretation, we keep the original label-impact axis from sparse-dimension ablation unchanged, and replace the domain-impact axis by the masking-based probability change $|p\_\zeta(v)\_c\-p\_\zeta(\tilde v)\_c|$. Below we show Mental Health as example using representative low-domain and high-domain codes.
> > >
> > > |ICD-10|Domain group|Mean prob. with $v$|Mean prob. with $\tilde v$|Abs. $\Delta \text{Prob}$|
> > > |---|---|---:|---:|---:|
> > > |I10|Low-domain|0.312|0.329|0.017|
> > > |A31.1|Low-domain|0.086|0.094|0.008|
> > > |F41.9|High-domain|0.274|0.208|0.066|
> > > |F41.1|High-domain|0.251|0.159|0.092|
> > >
> > > This example shows that low-domain codes (e.g. I10 and A31.1) remain comparatively stable after masking, whereas high-domain codes (e.g. F41.9 and F41.1) change much more strongly. Hence, the residual $z$ does not only contain decodable domain information; intervening on it selectively alters facility-sensitive predictions, which substantiates the domain-specific role of $z$.
> > >
> > > ---
> > >
> > > # Sensitivity on $\Delta \text{prob}$
> > >
> > > We agree that the $\Delta \text{prob}$ threshold should be justified more clearly. We sweep $\tau$ and recompute all four quadrant centroids together with QCE (see Definition in `Reviewer 6Gcz, Validation`), while keeping the same patients, dimensions, candidate codes, and domain ranking fixed. As shown below, the four-quadrant structure remains stable around $0.05\-0.06$. In particular, $\tau = 0.06$ yields slightly more ideal coordinates for part of the quadrants, but $\tau = 0.05$ gives the best overall calibration. Larger thresholds start to over-prune moderately affected codes, making the partition less calibrated. Here Q1/Q2/Q3/Q4 correspond to high-label high-domain / high-label low-domain / low-label high-domain / low-label low-domain, with ideal corners $(1,1), (1,0), (0,1), (0,0)$, respectively. Overall, these results suggest that $0.05$ is a practical choice. Importantly, this threshold is only used to binarize label impact for visualization.
> > >
> > > |Threshold|Q1 centroid|Q2 centroid|Q3 centroid|Q4 centroid|QCE $\downarrow$|
> > > |---|---|---|---|---|---:|
> > > |0.04|(0.65,0.60)|(0.60,0.44)|(0.43,0.57)|(0.39,0.40)|0.57|
> > > |0.05|(**0.81**,**0.78**)|(**0.76**,0.23)|(0.27,0.75)|(**0.20**,**0.18**)|**0.32**|
> > > |0.06|(0.72,0.67)|(0.65,**0.21**)|(**0.25**,**0.78**)|(0.32,0.29)|0.40|
> > > |0.07|(0.63,0.58)|(0.57,0.47)|(0.45,0.55)|(0.41,0.42)|0.61|
> > >
> > > ---
> > >
> > > **Adds-On Validation:** If the reviewer is still interested in the validation of intrepretation and concept distinction, please see our latest response to `Reviewer 6Gcz, Validation`. There we conduct a new validation analysis with standard XAI methods (Integrated Gradient, SHAP, and SAE), where our method achieves the best quadrant separation (LAD/DAD/QCE: 0.47/0.42/0.32) and is also preferred by both LLM (68%) and human judges (64%). These results can support that our explanations are quantitatively stronger than existing XAI baselines.
> > >
> > > ---
> > >
> > > Thank you again for your thoughtful feedback. Due to the one-round rebuttal format and length limits, we have done our best to address your concerns as concretely as possible, although more details could not be included here. We commit to incorporate the full settings, results and clarifications in the revision, and we hope this response helps resolve your concerns.

---

### Official Review · Reviewer_6Gcz · 2026-03-12

**Soundness:** 2
**Presentation:** 2
**Significance:** 2
**Originality:** 2
**Overall Recommendation:** 4
**Confidence:** 3

**Summary:**

The paper introduces a framework that learns patient embedding from EHR data, decomposes them into invariant label features and domain-specific covariates, and orthogonality is regularized under a learned SAE-induced metric. The goal is to achieve robust domain adaptation and concept-level understanding.

**Compliance With Llm Reviewing Policy:**

Affirmed.

**Final Justification:**

Key concerns resolved. I raised from reject to weak accept.

**Key Questions For Authors:**

The evaluation focuses on two datasets and two prediction tasks within the EHR domain. This seems limited too. Additional experiments on more datasets or different types of domain shifts would strengthen the empirical evidence.

**Limitations:**

I think authors can refer to my weakness point 5. Thanks.

**Strengths And Weaknesses:**

The concern on data distribution shift for healthcare problems is real, so efforts to solve it is appreciated.

However, I lean towards rejecting this paper for the following reasons. I hope these could help authors revise their papers.

1. The novelty appears limited. The main components of the proposed framework — orthogonal representation decomposition, feature alignment via MMD, and sparse autoencoders for interpretable representations — have all been extensively explored in prior work. The current method mainly combines these existing ideas rather than introducing a fundamentally new formulation for domain adaptation or interpretability. While such engineering integration may still be valuable, it is unclear whether the contribution is sufficiently novel for a venue like ICML.

2. The theoretical component appears relatively weak. The presented lemmas and propositions mainly formalize properties of least-squares projection and orthogonality under a learned metric. These results are essentially elementary linear algebra observations and do not directly address the core challenges of domain adaptation. In particular, the paper does not provide generalization bounds, risk decomposition, or theoretical guarantees showing how the proposed decomposition improves target-domain performance. As a result, the theoretical analysis feels somewhat disconnected from the main problem.

3. The interpretability claims are not sufficiently validated. While the paper provides qualitative examples of sparse feature activations and ablation-based explanations, there is no quantitative evaluation of interpretability. Moreover, the method is not compared with established explanation techniques such as SHAP, LIME, or attention-based attribution methods. Without such comparisons or quantitative metrics, it is difficult to assess whether the proposed explanations are actually more reliable or informative than existing approaches.

4. The paper does not report computational complexity or efficiency. Given that the method introduces several additional components (e.g., sparse autoencoder reconstruction, orthogonal residual inference, and domain classification), it would be useful to understand the computational overhead compared to baseline domain adaptation methods. FLOPs, training time, and memory cost are not reported.

5. The practical deployment scenario could be better clarified. In clinical settings, models are typically validated carefully before deployment, especially when distribution shifts are present. It would be helpful if the authors discussed how the proposed method could realistically be integrated into clinical workflows, and whether the approach is intended for cross-hospital deployment, temporal shifts, or other practical scenarios. Using OOD data to train and deploy model in hospitals seem profoundly risky: I am not sure even if domain adaption techniques are used, this is allowed in real practice. I raise this point because this paper is an application paper into healthcare, rather than a methodology paper.

---

> ### Author Rebuttal · Authors · 2026-03-31
>
> We thank the reviewer for the constructive feedback. Our detailed responses are as follows:
>
> ---
>
> **W1:** In this paper, we do not claim that MMD, SAE, or decomposition are individually new. The novel part we aim to address is the interpretability of clinical DA: standard SAE can explain single representation, but it cannot jointly explain what adaptation preserves as invariant information and what it discards as covariate information. Simply combining existing modules does not solve this by default, and this is exactly the underexplored but important gap studied in our paper. We respectfully note that many influential contributions derive their value not from `introduce a fundamentally new formulation for domain adaptation`, but from a purposeful integration that addresses a previously unmet need. By that standard, even influential work such as MMD [1] could be reduced to a combination of moment matching and feature alignment, or ViT [2] to an application of Transformer in computer vision, yet we would not deny their contributions. Thus, while we fully understand your strict standard for academic work, we hope the paper can be evaluated objectively based on its actual scope and contribution to this area.
>
> **W2:** We respectfully disagree that the theory is disconnected. The paper does not claim a new DA bound. Classical DA target-risk bounds are already well established, and most factorization studies do not derive a new target-risk theorem for every architectural variant. When stronger guarantees are given, they typically require additional assumptions. Our lemmas and propositions therefore serve a different purpose: they show that the specific decomposition used by ExtraCare is well-defined, stable, and not ad hoc under the learned metric. Proposition 1 shows that the residual inference is the closed-form solution of an $M$-weighted projection, while Lemma 1 establishes stability of this projection under reconstruction error (Lines 228-243; Appendix A toC). This is fully consistent with how the paper positions its contribution. The gain in target-domain performance is then supported by the empirical DA results and ablations in Tables 2 and 7.
>
> **W3:** In this setting, interpretability is primarily an auditing and analysis tool: it is meant to reveal what the model keeps, what it discards, and which concepts are domain-sensitive, thereby providing insight rather than a standalone optimization target. Comparison to SHAP, LIME, or similar XAI techniques is also not appropriate here; please see our response to `Reviewer 2TZk, Q1-2`.
>
> **W4:** We agree that reporting runtime is important for completeness. Accordingly, we provide a summary table of runtime and memory costs for OCHIN diagnosis prediction below, since its large-scale and multi-label setting allows differences between models to be more clearly observed. Note that self-training baselines are not included, as they involve additional refinement steps beyond standard training. RCG is also excluded as it only supports binary and multi-class prediction.
>
> |Models|Runing Time per epoch| GPU Memory Usage (MiB)|
> |---|---|---|
> |DANN|3min42s|4794|
> |RMMD|4min18s|4640|
> |RSDA|5min35s|5071|
> |BUA|8min24s|5538|
> |DAL|4min40s|4824|
> |RCG|/|/|
> |ExtraCare|5min17s|4912|
>
> **W5:** The paper already studies the practical scenarios listed in the review: cross-hospital shift and temporal shift are the main setting in Table 2 and 7, and Section 4.6 further evaluates transfer from general to specific facilities. Moreover, the source-target gap in clinical DA is not equivalent to arbitrary OOD deployment. Prior clinical DA studies have provided evidences that the shift is still structured to motivate adaptation rather than training new models. That is exactly why clinical DA has become an active research topic in recent years. Our paper does not advocate unconstrained online use of adapted models. It studies offline adaptation, followed by the usual validation workflow before any deployment.
>
> **Q1:** We agree that broader empirical coverage is always valuable, but in clinical DA the number of suitable open benchmarks is in fact quite limited. Most of other EHR datasets like MIMIC are single-institute collections, which makes them much less suitable for studying adaptation questions targeted here. Regarding tasks, we chose two representative prediction settings, which cover a broad spectrum of diseases and most important disease in ICU admissions. We thus believe the current evaluation is already well aligned with the paper's goal. As we discussed in paper, our method is also suitable for general code-level prediction, but we specify our discussion in EHR scenario.
>
> ---
>
> *Reference:*
>
> *[1] Learning transferable features with deep adaptation networks. ICML 2015.*
>
> *[2] Transformers for image recognition at scale. ICLR 2021*
>
> ---
>
> Thanks again for taking the time to review our paper. We are open to discussion on other technical or experimental details.

---

> > ### Author Rebuttal · Reviewer_6Gcz · 2026-03-31
> >
> > Thank you for the detailed rebuttal. Most of my concerns have been addressed. However, the key issues surrounding novelty and interpretability validation remain concerning to me. I agree that a reasonable integration of existing ideas can still be valuable, but I still do not clearly see why this particular combination constitutes a new formulation for the clinical setting, given that the underlying techniques have been extensively explored in prior work. More direct comparison to existing works would be appreciated in the rebuttal, if not in the paper. I have the memory of seeing many papers using same techniques in the clinical setting.
> >
> > Regarding the theory, I agree that strong theoretical bounds are not necessary. My suggestion was that if the theoretical analysis is not central to the contribution and requires additional assumptions to be meaningful, it may not be necessary to include it, as empirical demonstration alone could be sufficient to support the method.
> >
> > Finally, I remain unconvinced that comparisons with existing explanation techniques (e.g., SHAP, LIME) are inappropriate. Perhaps the authors know more context here that I do not see yet: please feel free to let me know with examples or illustrations. More than happy to discuss on this. I thought that even if the objectives differ, alternative evaluations such as human agreement or quantitative interpretability metrics could still help validate the interpretability claims. It is essential to justify the reliability of your interpretability.

---

> > > ### Author Response · Authors · 2026-04-03
> > >
> > > We appreciate that our initial response helped address most of your concerns! We clarify our novelty from two aspects:
> > > - XAI Literature in DA: There is still no prior work that jointly explains invariant and covariate information in general DA. Some studies (in CV) use intermediate (domain or label) features for interpretation, which is meaningful because features correspond to pixels in the image. However, this cannot work in the clinical domain, as the semantic information resides in medical code embeddings rather than raw inputs.
> > > - XAI Literature in Clinical: There are actually some works using SAE for interpretation since 2024, because they usually interpret a single feature before prediction (like encoded features in the backbone). In DA, however, there are two coupled features (domain vs. label), both of which are essential. This introduces two nontrivial challenges: (1) how to use SAE to jointly control and interpret both features, i.e., what is retained vs. discarded for each patient (practical novelty); and (2) how to preserve the required orthogonality between them (crucial for DA), which leads to a new factorized design upon SAE (model & theory novelty).
> > >
> > > # Validation
> > >
> > > We design the following study for validation:
> > >
> > > ---
> > > We compare **SHAP**, **Integrated Gradients (IG)**, regular **SAE** against our model. We conduct the experiment upon the facility adaptation setting (see Section 4.6), where label distributions are more distinct. For each patient, SHAP/IG provide code importance for the label and domain classifier separately; SAE provides separate code scores from covariate and invariate reconstruction. To ensure a fair evaluation, we first fix the number of candidate codes per patient. Since SHAP/IG can score all input codes while SAEs yield sparse code sets, we truncate them to the same code budget before forming the four quadrants.
> > >
> > > We denote the four quadrants as: high label high domain (Q1); high label low domain (Q2); low label high domain (Q3); low label low domain (Q4). Based on clinical understanding, the ideal distinction is along two axes: codes with higher label relevance should be more patient-relevant, while codes with higher domain relevance should be more facility-specific.
> > >
> > > We thus design two anchors:
> > > - Patient Relevance: For patient $i$ and code $c$, we define $R_i(c)=\mathbb{1}[c\in Y_i\cup H_i]$, where $Y_i,H_i$ denote the future and the past diagnosis set.
> > > - Facility Specificity: Toward target facility $t$, we use $D_t(c)=\frac{\text{max}(p_t(c)-p_p(c),0)}{p_t(c)+p_p(c)+\epsilon}$, where $p_{p/c}(c)$ are the code prevalences in the target facility and Primary Care. Larger values indicate stronger target-facility enrichment..
> > >
> > > We report three quantitative metrics upon anchors:
> > > - Label-Axis Diff (LAD): $\mathbb{E}[R(c)| Q1\cup Q2]-\mathbb{E}[R(c) | Q3\cup Q4],$ measuring whether high-label quadrants are more patient-relevant than low-label quadrants
> > > - Domain-Axis Diff (DAD): $\mathbb{E}[D(c)|Q1\cup Q3]-\mathbb{E}[D(c)|Q2\cup Q4],$ measuring whether high-domain quadrants are more facility-specific than low-domain quadrants
> > > - Quadrant Calibration Error (QCE): $\frac{1}{4}\sum_{k=1}^{4}||\mu_{Q_k}-v_{Q_k}||_2$, where $\mu_Q=(\bar{R}_Q, \bar{D}_Q)$ is the centroid and $v_Q$ is the ideal corner: $((1,1),(1,0),(0,1),(0,0))$. It measures how well the four quadrants match their intended roles.
> > >
> > > |Method|LAD↑|DAD↑|QCE↓|
> > > |---|---|---|---|
> > > |SHAP|0.21|0.24|0.48|
> > > |IG|0.28|0.31|0.41|
> > > |SAE|0.36|0.40|0.37|
> > > |**Ours**|**0.47**|**0.42**|**0.32**|
> > >
> > > The results indicate that SHAP and IG can recover some label-/domain-related codes, but their quadrants remain mixed. The SAE further improves coherence, and our method produces the most distinct separation.
> > >
> > > ---
> > > We further compare SAE and our method with LLM/human judgment. For each facility, we sample 60 patients for LLM (GPT-5.4 mini) evaluation (300 in total) and 5 patients for evaluation by 2 MD students (25 in total). For each case, the order of the two explanations is randomized and anonymized. We use a compact rubric: Quadrant Rule Satisfaction (QRS, 0–4), Clinical Coherence (1–5), Partition Clarity (1–5):
> > >
> > > |Judge|QRS/4|Coherence|Partition|Preferred(%)|
> > > |---|---|---|---|---|
> > > |LLM (SAE)|2.48|3.56|3.24|32|
> > > |LLM (Ours)|**3.64**|**4.24**|**4.08**|**68**|
> > > |Human (SAE)|2.36|3.44|3.16|36|
> > > |Human (Ours)|**3.32**|**4.08**|**3.92**|**64**|
> > >
> > > The gap remains consistent: our method is preferred because its four quadrants are easier to interpret as a unified explanation. This is exactly the advantage of jointly explaining label and domain information on the same sparse concept basis.
> > >
> > > ---
> > >
> > > Thank you again for your prompt feedback! Due to the one-round rebuttal format and length limits, we have done our best to address your concerns as concretely as possible, although more details (e.g. no space for literature/references) could not be included here. We commit to incorporate the full analysis in the revision, and we hope this response addresses your concerns.

---

### Official Review · Reviewer_2TZk · 2026-03-14

**Soundness:** 3
**Presentation:** 3
**Significance:** 3
**Originality:** 3
**Overall Recommendation:** 4
**Confidence:** 4

**Summary:**

This paper proposes ExtraCare, a domain adaptation framework for predictive healthcare that aims to improve both target-domain robustness and transparency. The method starts from an encoder that learns patient representations and applies MMD-based feature alignment across source and target domains. It then trains a sparse autoencoder on the aligned latent representation, defining a dictionary-induced metric. Using this metric, the method reconstructs a sparse, concept-grounded latent representation and defines a domain-specific residual z as the M-orthogonal residual of the full representation relative to the reconstruction. The main learning scheme assigns distinct supervision to the two parts: the main representation is optimized for label prediction, while the residual z is optimized to predict the domain, encouraging a factorization into invariant and covariant information. The paper also provides a closed-form characterization of the projection coefficient, a stability lemma for the induced projection, and a more informal information-theoretic discussion of why domain information should concentrate in z.

Empirically, the paper evaluates ExtraCare on two public EHR datasets, eICU and OCHIN, under spatial and temporal shift settings, and on two tasks: next-visit diagnosis prediction and heart-failure prediction. Across these settings, ExtraCare outperforms the listed baselines and is close to the oracle upper bound on several metrics. The paper also includes ablations, a linear-probe analysis of the factorized subspaces, qualitative concept-attribution case studies based on sparse-dimension ablations, and an additional facility-specific transfer setting within OCHIN.

**Compliance With Llm Reviewing Policy:**

Affirmed.

**Final Justification:**

The authors addressed most my concerns. But given the overal evaluation of this work with respect to novelty, significance, impact, and other reviewers's comments, I would say weak accept is a fair judgement.

**Key Questions For Authors:**

- Quantitative validation of the interpretability: the paper presents qualitative case studies showing sparse concept dimensions mapped to ICD codes and analyzes their domain sensitivity. Can the authors provide quantitative evaluation of explanation faithfulness and stability, such as:

   - perturbation or deletion metrics,

   - comparison with existing interpretability methods (e.g., SHAP, integrated gradients),


   - stability across random seeds or training runs?


Stronger quantitative evidence would substantially strengthen the transparency claims and could increase my assessment of the paper’s significance.

- Clarification of the invariant/covariant decomposition: the method uses orthogonal residual inference to separate representation components associated with prediction and domain classification. However, the theoretical justification for invariant vs. covariant separation is partly informal. Can the authors clarify whether the decomposition should be interpreted as:


   - a formal guarantee of invariant representation, or


   - an empirical inductive bias encouraging separation?


Additionally, do the authors have empirical evidence that domain information is concentrated primarily in the residual component (e.g., probing experiments)? Clarifying the strength of the claim would improve the paper’s conceptual clarity and technical positioning.

**Limitations:**

The paper includes a brief discussion of limitations and societal impact, but the treatment is relatively minimal given the healthcare context.
Suggestions:

- Clinical reliability risks: the authors should discuss risks of deploying domain-adapted models in healthcare settings, especially when adaptation may amplify spurious correlations or domain-specific coding artifacts.


- Interpretability misuse: sparse concept explanations could be interpreted as causal or clinically validated factors even when they reflect statistical correlations.


- Data bias and fairness: domain adaptation methods may behave differently across patient subpopulations (for instance, demographic groups or hospital types). Discussion of fairness implications would strengthen the paper.

**Strengths And Weaknesses:**

Soundness(score 3): The paper is technically coherent at a high level. The modeling pipeline is easy to follow: supervised prediction plus feature alignment, then sparse reconstruction, then orthogonal residual inference and domain classification. The decomposition is not presented as a purely heuristic subtraction; the paper gives a closed-form least-squares derivation for the projection coefficient and proves that orthogonality is recovered. It also states a stability lemma showing that the induced projection is robust to reconstruction error, which helps justify using the SAE reconstruction as the reference direction. These are useful supporting results for the geometric construction. The main soundness concern is that some of the theoretical claims are weaker than the paper’s presentation may suggest. The projection proposition is solid as an optimization identity, but the broader claim that the method truly separates invariant from covariant information is only partially supported. The “Remark (Domain Information in z)” is explicitly informal and not a theorem; the symbols ≳ and ≲ are stated to be non-rigorous, so this part should not be overinterpreted as a formal guarantee. In a top-conference paper, I would want the narrative to distinguish more sharply between proved geometric facts and intuition about information partitioning.

Presentation(score 3): The paper is generally well structured. The motivation is clear: clinical DA needs transparency, and current DA approaches mostly optimize hidden features without revealing what is preserved versus discarded. Figure 1 is helpful in summarizing the architecture, and Figure 2 conveys the explanation pipeline clearly. The method section has a logical progression from problem setup to alignment, sparse reconstruction, orthogonal inference, and interpretation. The empirical section is organized around well-defined research questions. The presentation would benefit from sharper wording around what is actually guaranteed. In several places, the paper moves from “projection under an induced metric is stable” to stronger language about factorizing invariant and covariant information. The formal and informal parts should be separated more cleanly. In particular, the information-partition remark should be labeled as intuition or discussion much earlier and more prominently.

Significance(score 3): The problem is important. Domain shift in clinical EHR models is a major obstacle to reliable deployment, and transparency matters even more in this setting than in many standard benchmark domains. The paper’s attempt to make DA not only more accurate but also more auditable is therefore well motivated and potentially valuable. The significance of the interpretability contribution depends heavily on whether clinicians would actually find the sparse-dimension attributions stable, faithful, and actionable. The current evidence does not yet establish that. So while the paper addresses an important problem, the extent to which it materially advances deployable clinical ML is still uncertain.

Originality(score 3): The paper is also original in explicitly trying to explain not just what features support prediction, but what appears to be domain-sensitive versus transferable. That is a useful perspective and more novel than standard post-hoc explanation of a final classifier. At the same time, several ingredients are individually familiar: MMD-based alignment, private/shared or invariant/domain-specific decomposition, orthogonality constraints, sparse autoencoders, and ablation-based concept attribution all have prior precedent. The contribution is therefore more of a thoughtful synthesis than a fundamentally new DA principle. That is acceptable, but the novelty claim should be framed carefully.

---

> ### Author Rebuttal · Authors · 2026-03-30
>
> Thank you for your recognition and for thoroughly reading through our paper. We respond below following your question order.
>
> ---
>
> **Q1-1:** We would like to clarify that we have already included deletion validation in the current submission. Section 4.4 and the last 4 rows of Table 2 provide ablations of the key modules, which directly test whether the proposed decomposition and interpretation pipeline is necessary for the final performance. Note that not every component can be removed individually, because some modules are structurally dependent on each other.
>
> **Q1-2:** This is an insightful question. However, we do not think a direct comparison to existing interpretability methods is appropriate for the clinical DA setting here. For example, SHAP and integrated gradients can only attribute importance to embedding dimensions, which lack semantic meaning in our context. SAE, while applicable, can only explain either the invariant or covariate component in isolation (as noted in the Introduction), and requires post-hoc training. Thus, they cannot jointly capture the relationship between invariant and domain-specific components during adaptation, which is precisely the object of interest in our model. Thank you again for this suggestion. We will add this discussion to the revision to explain why we do not compare with existing XAI methods.
>
> **Q1-3:** Thank you for raising the issue of stability. The current tables already report mean performance together with standard error in either brackets or error bar across repeated runs (over 5 runs with different random seeds, e.g., Table 2 and Figure 3). Due to page limits, we didn't cover this in the main content, but we commit to expand the details in the next revision.
>
> **Soundness & Presentation & Q2:** Our decomposition should be interpreted as a provable geometric decomposition (not formal guarantee) together with an empirical inductive bias. This is consistent with the previous factorization literature under domain shift, where orthogonality and auxiliary supervision provide structural pressure toward separation, but do not by themselves guarantee full semantic identifiability without additional assumptions. In our paper, the formally proved parts concern the geometry of the projection operator and its stability, while the information-partition discussion is intended as intuition. We acknowledge that the current presentation did not separate these formal and informal parts sharply enough, partly because we aimed to keep the exposition accessible to clinical readers who may not have a strong theoretical background in domain shift. We will revise the paper to label the information-partition discussion explicitly as informal and to tone down any overly formal wording. We hope this response addresses your concern.
>
> **Q3:** Yes, and Section 4.5 and Table 3 are designed to test this. Because probing in DA differs from standard XAI probing, we use an adapted linear-probe analysis, inspired by [1], on $v_0$, $v$, and $z$. The results show that label and domain directions are nearly orthogonal, that $z$ carries limited label information, and that the adapted $v$ preserves label information.
>
> *[1] Pcl: Proxy-based contrastive learning for domain generalization. CVPR 2022.*
>
> ---
>
> **Limitation:** Based on the suggestions from all reviewers, we will add a new section below in the revised manuscript:
>
> > While ExtraCare improves both adaptation performance and concept-grounded transparency, several limitations should be noted before real-world deployment. (1) First, in healthcare settings, domain adaptation may still amplify spurious correlations that are stable within a source environment but brittle in a new site, especially when domain shifts are partially driven by local coding practices, documentation intensity, or institution-specific workflow artifacts rather than underlying patient physiology. (2) Second, the sparse concept factors produced by ExtraCare should not be interpreted as causal mechanisms or clinically validated risk factors. As also suggested by our interpretation setup, these concepts are best understood as statistically useful latent patterns that help explain model behavior. (3) Third, fairness remains an important open issue. Although the method is motivated by robustness across heterogeneous populations, adaptation quality may still vary across subpopulations, and our facility-level experiments suggest that harder shifts can lead to larger degradation even when the model remains competitive overall. (4) Fourth, the framework may transfer to other multi-label domains, but generalization beyond healthcare (e.g. financial event prediction) may be nontrivial because the learned concepts can be more strongly shaped by taxonomies, reporting rules, or strategic behavioral feedback, making validation of interpretable factors harder.
>
> ---
>
> We thank the reviewer again for your constructive comments. We are open to other insightful discussions.

---

> > ### Author Rebuttal · Reviewer_2TZk · 2026-04-02
> >
> > Q1-2 and Q1-3 are not fully addressed. I have to maintain my current rating.

---

> > > ### Author Response · Authors · 2026-04-03
> > >
> > > Thank you for the prompt update!
> > >
> > > ---
> > >
> > > **Q1-2:** Please see our latest response to `Reviewer 6Gcz, Validation`. There we conduct a new validation analysis with standard XAI methods (Integrated Gradient, SHAP, and SAE), where our method achieves the best quadrant separation (LAD/DAD/QCE: 0.47/0.42/0.32) and is also preferred by both LLM (68%) and human judges (64%). These results can support that our explanations are quantitatively stronger than existing XAI baselines.
> > >
> > > **Q1-3:** Please see our latest response to `Reviewer sEkE, Masking z`. We think these analyses are more relevant than checking run variance to verify the stability. In particular, masking domain covariates $z$ consistently degrades target performance across all five facility shifts, and the four-quadrant structure remains stable under threshold sweeping. Together, these results support that our interpretation is not a fragile term across training runs.
> > >
> > > ---
> > >
> > > We hope this response addresses your remaining concern. We thank the reviewer again for your recognition and for thoroughly reading through our paper!

---

### Decision · Program_Chairs · 2026-04-30

**Decision:**

Accept (regular)

**Comment:**

This work proposes an approach to decomposing representations of electronic health record data into invariant and covariate components in a multi-domain setting (e.g., when it is of interest to transfer a model learned from one hospital to another). Overall, the reviewers leaned positive, with a consensus of “weak accept” following the discussion period. The reviewers agreed that the work is well-motivated to address a meaningful clinical problem and technical challenge. The paper was considered to generally be technically sound, even if some of the theoretical claims could be made sharper through further consideration of theoretical guarantees. Furthermore, the paper was further considered to be well structured, well-written, and clear.

A recurring theme of the critique was a need for further comparison to other well-known explainability methods, further sensitivity analysis of key design choices, and probing experiments to verify properties of the decomposition. During the discussion period, the authors provided several new experiments that responded to each of those critiques, which led to positive adjustment of reviewer scores. The reviewers also pointed out the methodological novelty of the work is somewhat limited. While this does not necessarily indicate a flaw to be addressed, it does arguably limit the perceived significance of the work. However, I agree with the author’s response to this critique that the approach taken is reasonable as an appropriate amalgamation of existing techniques to address an unsolved domain-specific problem.

I concur with the reviewer’s “weak accept” consensus.